# In-situ resonant band engineering of solution-processed semiconductors generates high performance n-type thermoelectric nano-inks

Ayaskanta Sahu [1,2], Boris Russ[2,3], Miao Liu [4,5], Fan Yang[2,6,7], Edmond W. Zaia[2,3], Madeleine P. Gordon[2,8], Jason D. Forster [2], Ya-Qian Zhang[9,10], Mary C. Scott[9,10], Kristin A. Persson [5,13], Nelson E. Coates[2,11], Rachel A. Segalman [12] & Jeffrey J. Urban [2✉]

Thermoelectric devices possess enormous potential to reshape the global energy landscape by converting waste heat into electricity, yet their commercial implementation has been limited by their high cost to output power ratio. No single "champion" thermoelectric material exists due to a broad range of material-dependent thermal and electrical property optimization challenges. While the advent of nanostructuring provided a general design paradigm for reducing material thermal conductivities, there exists no analogous strategy for homogeneous, precise doping of materials. Here, we demonstrate a nanoscale interface-engineering approach that harnesses the large chemically accessible surface areas of nanomaterials to yield massive, finely-controlled, and stable changes in the Seebeck coefficient, switching a poor nonconventional *p*-type thermoelectric material, tellurium, into a robust *n*-type material exhibiting stable properties over months of testing. These remodeled, *n*-type nanowires display extremely high power factors ($\sim$500 $\mu$W m$^{-1}$K$^{-2}$) that are orders of magnitude higher than their bulk *p*-type counterparts.

[1] Department of Chemical and Biomolecular Engineering, New York University, 6 Metrotech Center, Brooklyn, NY 11201, USA. [2] The Molecular Foundry, Lawrence Berkeley National Lab, 1 Cyclotron Road, Berkeley, CA 94720, USA. [3] Department of Chemical and Biomolecular Engineering, University of California Berkeley, 201 Gilman Hall, Berkeley, CA 94720, USA. [4] Institute of Physics, Chinese Academy of Sciences, No.8 3rd South Street, Zhongguancun, Haidian District, Beijing 100190, P.R. China. [5] Energy Technologies Area, Lawrence Berkeley National Laboratory, 1 Cyclotron Road, Berkeley, CA 94720, USA. [6] Department of Mechanical Engineering, Stevens Institute of Technology, 1 Castle Point Terrace, Hoboken, NJ 07030, USA. [7] Department of Mechanical Engineering, University of California Berkeley, 6141 Etcheverry Hall, Berkeley, CA 94720, USA. [8] Applied Science and Technology Graduate Group, University of California Berkeley, 210 Hearst Memorial Mining Building, Berkeley, CA 94720, USA. [9] Department of Materials Science and Engineering, University of California Berkeley, 2607 Hearst Ave, Berkeley, CA 94720, USA. [10] National Center for Electron Microscopy, Molecular Foundry, Lawrence Berkeley National Laboratory, 1 Cyclotron Road, Berkeley, CA, USA. [11] Department of Physics, University of Portland, 5000 N. Willamette Blvd., Portland, OR 97203, USA. [12] Departments of Chemical Engineering and Materials, University of California Santa Barbara, Engineering II Building, Santa Barbara, CA 93106, USA. [13] Present address: Department of Materials Science and Engineering, University of California Berkeley, 2607 Hearst Ave, Berkeley, CA 94720, USA. ✉email: jjurban@lbl.gov

With the worldwide demand for energy and thermal management rapidly accelerating, thermoelectric (TE) devices, which can convert thermal energy to electrical energy (and vice versa), are receiving increasing attention for power generation, waste heat recovery, and solid-state cooling[1,2]. The current advent of ubiquitous internet-of-things such as medical devices, sensors, integrated, wearable, and flexible electronics require mW-level power sources that are adaptable to a wide genre of substrates and of arbitrary geometry[3–5]. Mechanically flexible body-powered thermoelectric generators (TEGs) coupled with appropriate heat exchangers are a promising class of devices that can cater to these emerging portfolio of applications through robust high-performance solution-processed materials compatible and conformable to various surfaces, substrates, and geometries with arbitrary form factors[4,6]. Typical processing techniques used with traditional bulk polycrystalline TEG materials such as solid-state oxides and chalcogenides (i.e., metal-oxide chemical vapor deposition) require high temperatures and generate rigid form factors and often brittle devices. Few pixelated and paneled devices have been attempted with these bulk TEs albeit at the cost of several additional processing steps[7,8]. Flexible TEGs require both a solution-synthesized nanomaterial (for precise size control and efficient phonon scattering) and solution-processable (for conformal fabrication) material—a need which cannot be met by bulk TEs or energy-intensive spark plasma sintered TEs made from nanomaterials[9,10]. In general, true solution processing compatible with scalable roll-to-roll fabrication has been confined to soft organic TEs with a couple of hybrid organic–inorganic alternatives.

The efficiency of a TE material is determined by a dimensionless figure-of-merit, $ZT = S^2\sigma T/\kappa$, where $S$ denotes the thermopower (or Seebeck coefficient), $\sigma$ the electrical conductivity, $\kappa$ the thermal conductivity, and $T$ the absolute temperature. Due to the inherent coupling of the parameters $S$, $\sigma$, and $\kappa$, designing high $ZT$ materials remains an ongoing challenge[11–14]. While significant advances have been made in individual $p$-type organic materials such as PEDOT:PSS (power factors, $S^2\sigma$ ~700 μW m$^{-1}$ K$^{-2}$) and $n$-type hybrid TiS$_2$-C$_{60}$ nanomaterials ($S^2\sigma$ at 400 K ~ μW m$^{-1}$ K$^{-2}$)[9,15,16], the former relies on expensive ionic liquids and multi-stage chemical treatments with PEDOT:PSS being moisture sensitive and the latter requires multistep energy-intensive fabrication at low pressures and high temperature for days followed by solid-state milling[17]. While I$_2$-doped polyaniline can exhibit high $p$-type power factors (1350 μW m$^{-1}$K$^{-2}$), it is practically unusable due to its poor stability and processability[18]. In the same vein, the best-in-class $n$-type solution-processed materials comprise single-walled carbon nanotubes, halogenated benzodifurandione-based oligo ($p$-phenylenevinylene) (BDPPV) derivatives, and perylenedimides with power factors around 20–30 μW m$^{-1}$ K$^{-2}$ [19–21]. Plagued by some of these aforementioned massive shortcomings, there exist no viable solution-processed options for environmentally stable long-term TE performance.

Here we discuss a general strategy for transforming materials that have never been considered suitable for TEs into promising candidates that compete with state-of-the-art $n$-type solution-processed TE materials. To achieve this, we employ an innovative approach for precision in-situ doping through covalent atomic resurfacing of nanoscale materials, which we leverage to create band-engineered TE materials. Stable, controllable doping is a central challenge in materials science, pertinent to nearly every field of optoelectronics and energy materials. Yet there has been little innovation in this area—doping strategies still predominantly rely on introduction of an aliovalent element into the lattice via energy-intensive annealing, which is spatially inhomogenous, often introduces undesirable scattering mechanisms, and does not always translate to nanomaterials well. Our methodology is based upon true band conversion or band tuning of materials resulting from nanoscale chemical resurfacing, which effectively dopes the materials, opening up alternative strategies for materials processing, and enabling facile development and implementation of tailored module architectures[1].

## Results

**Band engineering methodology through surface remodification.** While nanostructured TE materials have historically capitalized on enhanced boundary scattering and attendant reduction of $\kappa$ for improved performance[22–31], the large chemically accessible interface inherent to all nanomaterials has proven, in many cases, to be a net detraction due to enhanced boundary scattering of electrical carriers[31]. However, reimagined, it can provide a powerful knob for manipulating electronic TE properties ($S$ and $\sigma$)[32,33]. Bound ligands, tailored through solution-processed ligand exchange[34], can potentially enable doping of the parent nanomaterials by charge transfer at the interface or by hybridizing and modifying local densities of states (DOS)[33,35–37]. Such solution surface modification can be coupled with colloidal bottom-up synthesis of nanomaterials in a low-cost and high-throughput commercially viable process yielding dimensionally and chemically precise structures[38,39]. Reports utilizing solution-processed nanomaterials without additional energy-intensive (and costly) post-processing, such as spark plasma sintering (SPS), are limited[40–46]. In contrast, we use a surface modulation approach to surface-dope tellurium (Te) nanowires (NWs), a poor $p$-type TE material, with small molecules in solution. At high doping concentrations, we obtain $n$-type Te NWs from a material system that was formerly $p$-type. By precisely controlling the doping level, we can monotonically shift the Fermi level (E$_F$) of films cast from these Te NWs with fine gradation (spanning a range of ~1000 μV/K with steps of about tens of μV/K in Seebeck coefficient) from $p$-type to $n$-type behavior. Finally, we demonstrate that these NW films are stable over days under constant operation and can be integrated into flexible substrates with both the $p$- and $n$-legs fabricated via solution processing. The key development is a scalable and materials-agnostic interface-engineering technique that harnesses the large, chemically accessible surface areas of nanomaterials for both phonon scattering as well as precisely manipulating the electronic properties.

In contrast to traditional electronic doping strategies that employ aliovalent atoms (e.g., boron or phosphorus in silicon (Si)) in order to introduce extra charge carriers, another parallel approach to improve the electronic properties of TEs has been to manipulate the local DOS (resonant levels) around the Fermi level of the host material (e.g., Tl, Cr, and Ti in PbTe, Al in PbSe, Sn in Bi$_2$Te$_3$, In in SnTe etc.)[35,47–53]. In order to achieve significant density of transport channels, dopant concentrations need to be high. Unfortunately, at high impurity concentrations, traditional bulk materials run into issues of dopant phase segregation and intense carrier scattering. Our method uses the large surface area provided by the nanomaterials to achieve high doping concentrations limiting the carrier scattering to the surface (which would otherwise anyway exist in all nanostructured bulk TEs) akin to modulation doping. To the best of our knowledge, this is the first report of a doping methodology which uses isoelectronic dopants to introduce resonant DOS to such a high degree that inverts the carrier nature in a degenerately doped nanocrystalline semiconductor. All prior work focused on choosing a high-performance TE material and then trying to improve its power factors by using band engineering around the Fermi level—improving the $p$-type (or $n$-type) performance of a $p$-type (or $n$-type) material. Our nonconventional approach aims

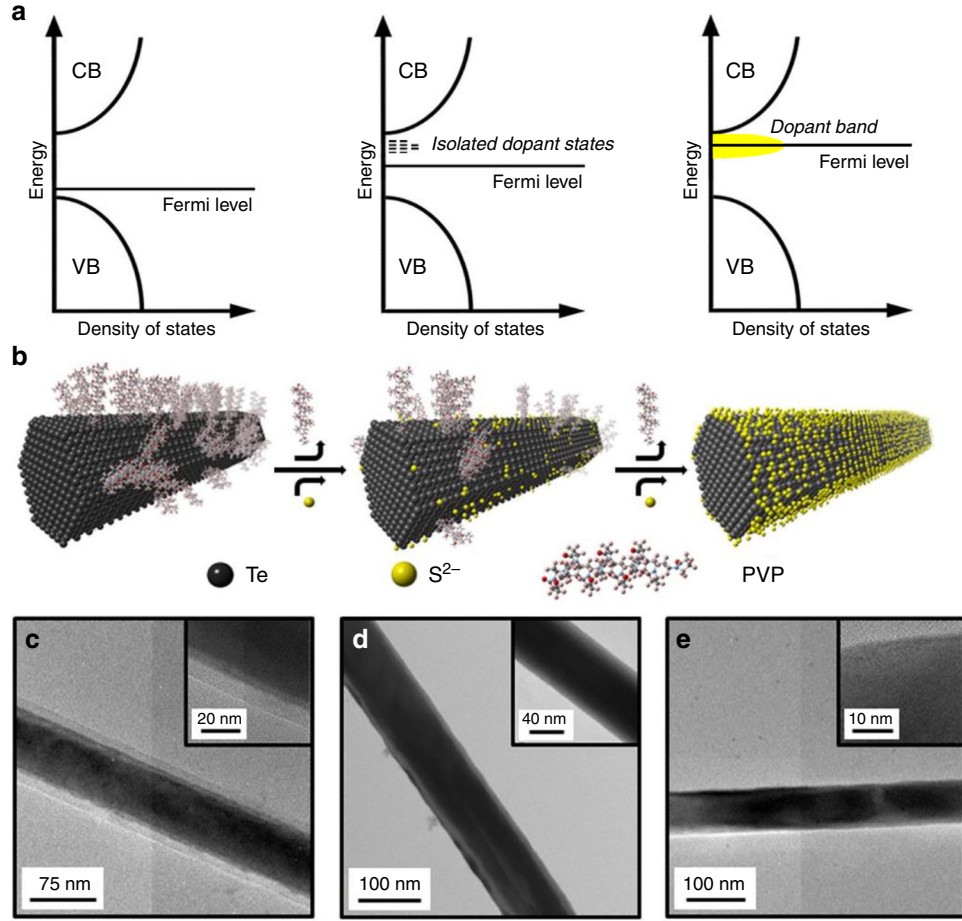

**Fig. 1 Resonant band engineering by interfacial resurfacing. a**, **b** Schematic of the doping process (not to scale) showing the evolution of the density of states and the change in Fermi level with removal of the polymer (polyvinylpyrrolidone, PVP) and attachment of the sulfur ($S^{2-}$) atoms. CB and VB stand for conduction band and valence band, respectively. At low sulfur concentration, isolated states originate close to the CB of tellurium which transform to a prominent sulfur-generated dopant band at high concentrations. **c–e** Transmission electron micrographs of undoped, intermediate- and heavily doped 80-nm diameter tellurium (Te) nanowires (NWs), respectively. The insets show high resolution images with both the inorganic component and the polymer layer. While one can observe a thick polymer layer in **c**, the doped sample in **d** shows a thinner layer and the one in **e** shows no evidence of any polymer.

to break free from this traditional approach by choosing an unimpressive $p$-type TE i.e., Te ($S^2\sigma$ ~1–10 µW µW m$^{-1}$ K$^{-2}$) and transforming it into a high performance $n$-type TE ($S^2\sigma$ ~500 µW m$^{-1}$ K$^{-2}$). For the first time, we report $n$-type transport in Te —traditionally a $p$-type material. No prior work—neither in the thin-film community nor the bulk community—has observed $n$-type transport in this system[54–56]. This would be tantamount to inverting the carrier type of gold or copper and making $p$-$n$ logic elements out of that single material alone. This work shows the potential that doped nanomaterials offer to give rise to interesting electronic properties not extant in the corresponding bulk system. More generally, these results suggest that interface engineering of nanostructures holds great potential for driving forth the next generation of hybrid materials for applications in $p$-$n$ based logic, transistors, optoelectronics, and TEs.

We have developed a surprisingly facile method for atomically resurfacing TE nanomaterials (Fig. 1) and use a simple recipe to dope our Te NWs with $S^{2-}$. We synthesize ~80-nm diameter Te NWs with a capping polymer (polyvinylpyrrolidone (PVP)) following established protocols[43,44] and then mix the resulting NWs dispersed in deionized water with a metal sulfide salt, $Na_2S$ in this case. We tune the doping concentration simply by changing the concentration of $S^{2-}$ in solution. The doped NWs retain their solution processability and are stable for months under ambient conditions. Figure 1b shows a schematic of this

process. We postulate that the $S^{2-}$ ions first penetrate the polymer coating on the as-synthesized NWs and attach to any unpassivated surface Te atoms. Increasing the dopant concentration gradually expulses the polymer coating, thus creating more surface sites for $S^{2-}$ attachment. Finally, in the heavily doped limit, almost all the polymer is displaced and the NW surface is fully coated with $S^{2-}$ atoms. Transmission electron micrographs (TEM) show that the NWs retain their size and crystallinity even at the highest doping concentration (Fig. 1c–e and Supplementary Fig. 1).

**Morphology of nanowires pre- and post-surface modification.** To analyze the polymer concentration on our Te NWs, we performed thermogravimetric analyses (TGA) on various samples with different dopant concentrations (Supplementary Fig. 2). We observe a significant and monotonic loss in polymer mass as a function of increased doping which corroborates well with the TEM results and our assumption that $S^{2-}$ dopants slowly displace the polymer. X-ray diffraction (XRD) patterns (Supplementary Fig. 2) also show no significant changes in Te structure as a function of dopant concentration. A unique and distinctive advantage of our approach over other chemical doping methods for nanomaterials is the absence of any Te oxidation in the doped samples even after weeks of exposure to ambient conditions. As-

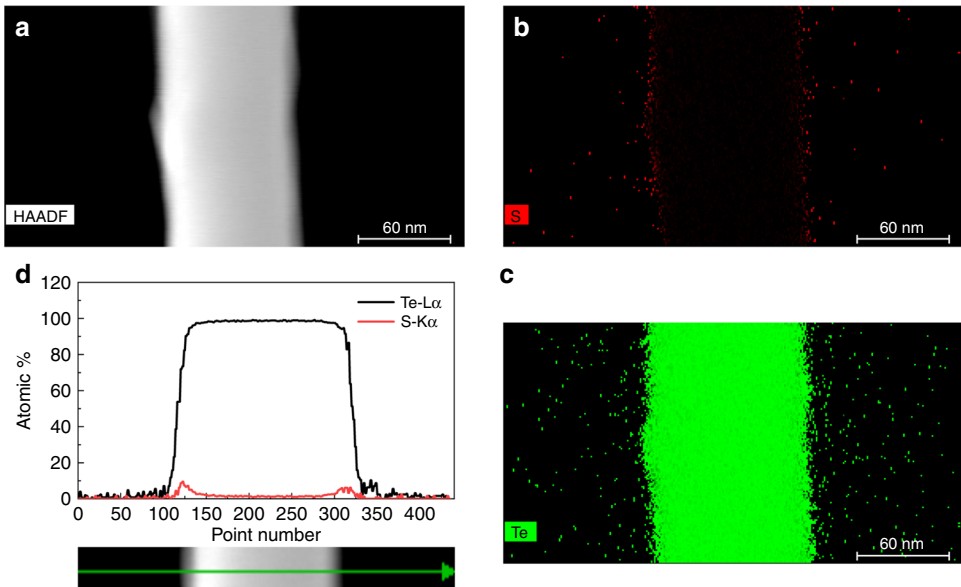

**Fig. 2 Structural characterization of dopant location. a** High-angle annular dark-field scanning transmission electron microscopy (HAADF-STEM) and **b**, **c** energy-dispersive X-ray (EDX) maps of sulfur and tellurium, respectively, in a sulfur-doped tellurium nanowire showing the presence of minute amounts of sulfur in the sample **d** line scans demonstrating that sulfur atoms are primarily concentrated on the surface of the nanowire.

synthesized Te NWs do oxidize within the same time frame (black curve in Supplementary Figs. 2, 3). This proves that even in the extremely lightly doped limit, the $S^{2-}$ dopants specifically target and serve to passivate any vacant surface Te site which might be prone to oxidation and provide remarkable stability. In order to prove that the sulfur exists only on the surface of the Te NWs and does not diffuse into the bulk (which would be akin to bulk doping) rather than surface engineering, we perform high-angle annular dark-field scanning transmission electron microscopy (HAADF-STEM) and energy-dispersive X-ray (EDX) maps on our samples (Fig. 2 and Supplementary Fig. 4). STEM-EDX and EDX line scans were used to analyze the elemental distribution in S-doped Te NWs. The images clearly show that sulfur is concentrated in the surface region, while Te is concentrated in the core region.

**Thermoelectric properties**. The excellent dispersability of our materials (Supplementary Fig. 5, Zeta Potential Measurements, Supplementary Figs. 6–10) allows us to cast smooth films on various substrates; Supplementary Fig. 5 shows scanning electron microscopy (SEM) images of drop-cast films of NWs from water. Even though the films are porous, they form a dense interconnected network of NWs. While the doped Te NW films (~2.4% atomic concentration) show no apparent change in structure or morphology, we observe dramatic electronic changes in the TE properties of these wires. While the thermopower of as-synthesized Te NWs is positive (Supplementary Fig. 11a), it switches sign and is negative for the doped NWs (Supplementary Fig. 11b). The sign of the thermopower of a material is a reliable indication of the nature of the majority charge carriers in that material, and the incorporation of the surface $S^{2-}$ dopant unambiguously switches Te from $p$-type to $n$-type.

**Density functional theory calculations**. Intuitively, one might expect the negatively charged surface S atoms to donate electrons to Te NWs to dope them $n$-type. In general, however, predicting the effect of doping is not trivial since, for example, deep defect levels in the host material might scavenge these carriers and remove them from the conduction band[57]. Additional issues arise

with nanoscale materials since undesirable redox reactions will occur more readily in ambient conditions due to the larger surface energy relative to bulk crystals[58,59]. Therefore, in order to gain a greater understanding of the doping mechanisms at work with these surface ions, we examined the charge-transfer effect between S adatoms and Te on the (010) surface since it is the most stable and exposed surface (Fig. 3a) using density functional theory (DFT) calculations (details in "Methods," Supplementary Figs. 12–17 and Supplementary Note 2)[60–63]. To gain insight as to how electron doping is induced by sulfur adsorption, we calculate the charge-transfer effect directly between sulfur and Te based on two extreme cases: physical adsorption from weak van-der-Waals forces as shown in Fig. 3b and chemical adsorption with strong bonding displayed in Fig. 3c. Structures are relaxed fully to equilibrium configurations while maintaining the sulfur atoms at fixed positions. In both cases, the charges are redistributed between the S adatoms and the Te lattice, as shown in Fig. 3b, c, and form localized dipole moments at the Te/S interface with a penetration depth of ~3–4 Te atomic layers in general. By integrating charge transfer in the in-plane direction of the film, the amount of charge redistribution between S adatoms and the Te slab is extracted along the surface normal direction. The results indicate that sulfur attracts electrons more prominently while Te loses electrons and becomes slightly positively charged, resulting in a negatively charged surface region around the surface-bound S atoms. This charge redistribution behavior can be understood by the higher electronegativity of sulfur compared with Te, which results in sulfur generally exhibiting a stronger tendency for attracting electrons than Te.

The explanation is further supported by X-ray photoelectron spectroscopy (XPS) measurements (Supplementary Figs. 18–20). A slight blue shift can be observed in the Te peaks in the doped samples which proves that Te atoms are marginally more $p$-type doped in the S-doped samples (Supplementary Fig. 20). While isovalent dopants are not perceived as traditional dopants, recent studies have shown that it is indeed possible to dope a material with a corresponding element exhibiting the same valence state. These results are in good agreement with the experimental observations and confirm the proposed mechanism that sulfur adsorption can indeed generate a hybrid $n$-type Te-sulfur

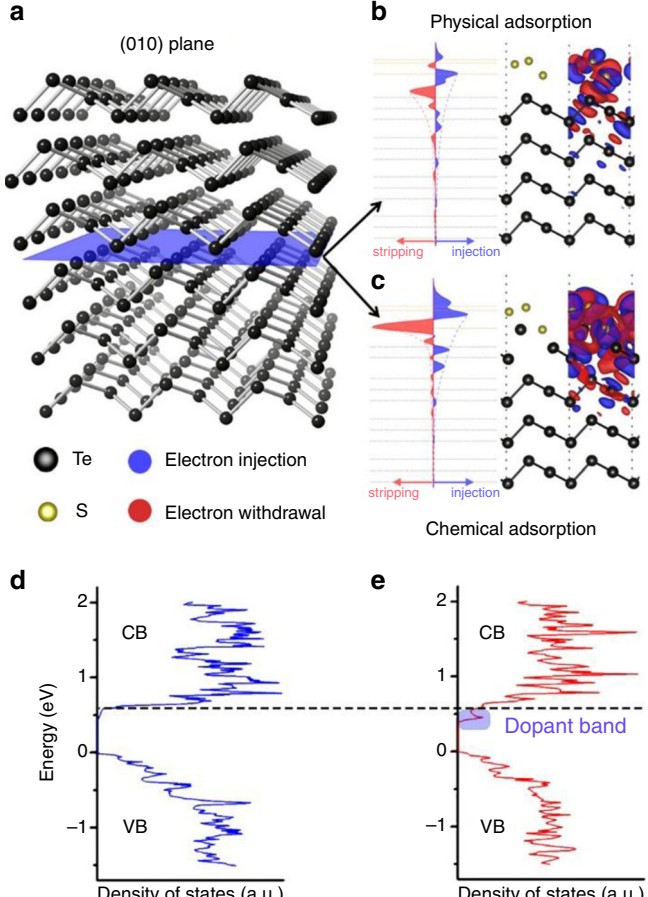

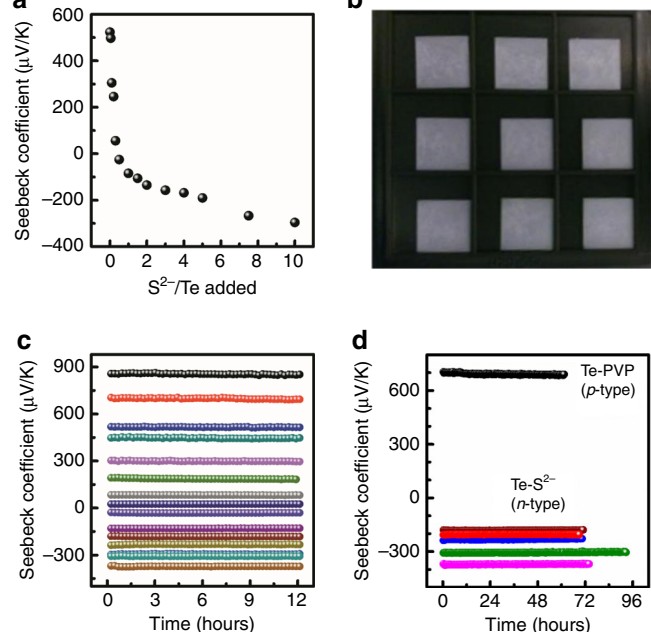

**Fig. 4 Tuning Seebeck coefficients with controlled doping. a** Seebeck coefficient from a series of doped Te NW samples versus the amount of $S^{2-}$ added to the exchange solution, normalized to the total number of Te atoms present on the surface of the NW. **b** Representative set of smooth drop-cast films used for the measurements. **c** Short-term stability tests from a series of Te NW samples with varying doping concentrations. **d** Long-term stability tests for as-synthesized $p$-type Te NWs capped with polyvinylpyrrolidone (black) and multiple batches of fully surface exchanged Te NWs with $S^{2-}$ (colors). The data in green is from a sample stored in ambient for nearly 7 months and the data in red is from a device stored in the glove box for 23 months.

**Fig. 3 Charge transfer and evolution of a resonant dopant band with sulfur doping. a** Illustration of the atomistic structure and (010) surface, of the hexagonal tellurium nanowire using density functional theory calculations. The supercell configuration and the charge-transfer effect for **b** physical adsorption and **c** chemical adsorption of equal amount of sulfur adatoms on tellurium along (010) surface. The blue isosurface represents injection of electrons, and red for stripping or withdrawal of electrons. For clarity, the iso-charge contours are only shown in the right halves of **b** and **c**. Plane-integrated charge transfer along the surface normal direction is shown for both cases and is plotted with the same scale, so that they are quantitatively comparable. **d**, **e** Calculated surface density of states using density functional theory (DFT) for tellurium ($p$-type) and sulfur-doped tellurium ($n$-type) with CB and VB referring to the conduction and valence bands of bulk tellurium, respectively. The dotted black line denotes the edge of the CB. A new dopant band (shaded ellipse) emerges close to the conduction band edge for the sulfur-doped tellurium.

nanostructure. From the transport measurements wherein we observe $n$-type Te, it is evident that the $E_F$ ought to be close to the conduction band edge. Ultraviolet photoelectron spectroscopy (UPS) measurements were conducted to measure the work function (WF) in Te and Te–$S^{2-}$ thin-film samples (Supplementary Figs. 21, 22). The measurements qualitatively demonstrate a shift toward lower levels of WF post doping (from 4.0 to 3.7 eV) which, in turn, indicates a shift of the $E_F$ from close to the valence band edge of $p$-type Te to the conduction band for $n$-type Te.

Controlling the charge carrier concentration and in turn the location of the $E_F$ in a TE material is essential to obtaining high $ZT$ values since the value of the Seebeck coefficient is determined by the band structure around $E_F$. Typical materials used for TE applications are degenerately doped semiconductors, thus introducing a trace quantity of dopants is not sufficient to move

the $E_F$ enough to maximize $ZT$. Therefore, to optimize $ZT$ in traditional TEs, a semiconductor is doped with nearly 1% dopants and then it is alloyed heavily to reduce lattice thermal conductivities [e.g. $(Bi_{1-x}Sb_x)_2Te_3$, $x$ ~0.1–0.2]. However, this alloying procedure can potentially introduce drastic changes in the band structure of the host material, and the optimal $E_F$ of the alloy is often different from that of the parent semiconductor material. An alternative approach to improving TE properties demonstrated by Heremans et al. utilizes dopants to introduce a resonant DOS within the band which allows one to simultaneously increase both $S$ and $\sigma$[35]. However, this approach relies on the fact that the $E_F$ be within a narrow energetic window close to that sharp DOS which is not guaranteed a priori. In addition, finding appropriate dopant-host combinations is also challenging, and only a handful of cases have been demonstrated[35,47–53]. To elucidate the doping mechanism, we conduct extensive DFT studies to calculate the band structure and DOS for both the doped and undoped systems (Supplementary Note 2, Supplementary Figs. 12–17). The calculated DOS show the presence of a new dopant band close to the conduction band of Te in the sulfur-doped samples (Fig. 3d, e) somewhat resembling the observations by Heremans et al. All previous reports on using resonant DOS in bulk semiconductors focus on improving the Seebeck coefficient of the host material while retaining the majority carrier type. In contrast, our approach completely inverts the majority carrier type from holes to electrons by tweaking the band structure. Also, our approach of surface doping a nanoparticle relies on a charge-transfer mechanism similar to modulation doping which isolates dopants from the

medium of transport[33,36], thereby reducing ionized impurity scattering, and allowing us to preserve the semiconductor band structure, as well as controllably move the $E_F$ to an optimal level.

**Tunability and stability of thermopower.** An inherent advantage of our surface doping technique is that it automatically generates a control sample for tracking changes due to doping. In other approaches, where the impurities are generally added during the growth of the nanostructure, it remains challenging to synthesize samples that differ only in dopant concentration, without disrupting the size, shape, or crystallinity of the samples. However, a significant strength of our approach is that we can prepare an entire series of samples that only differ by the amount of $S^{2-}$ added. We note that the size, morphology, and crystallinity of the wires (Fig. 1c–e and Supplementary Figs. 1, 2) remain otherwise identical—another unique feature of this approach relative to substitutional doping where undesirable ionized impurity scattering can be enhanced as a by-product of doping. In Fig. 4a, we plot the Seebeck coefficient of Te NWs (measured from neat films cast on glass substrates—Fig. 4b) versus the amount of $S^{2-}$ added in the initial solution and observe a consistent monotonic decrease in the Seebeck coefficient with increasing $S^{2-}$ addition. This trend is apparent across samples from different synthetic batches, which points toward a mechanism of controlled $S^{2-}$ incorporation.

The monotonic decrease of the value of the Seebeck coefficient from positive to negative is consistent with the view that we shift the location of the $E_F$ in the film away from the valence band toward the conduction band, slowly transforming Te from a p-type to an n-type material. While the exact origin of these effects is not yet clear, we propose that the introduction of additional states by the sulfur atoms leading to a dopant band combined with extra electrons filling the lowest energy states in the NW can lead to these observations. Assuming bulk carrier concentrations of $4.8 \times 10^{17}$ holes/cm³ for Te gives nearly $10^4$ holes/Te-NW (~80-nm diameter)[55]. A simple calculation supports this notion, showing that even at 2.5% concentration (Supplementary Note 3 and Supplementary Fig. 18), surface-bound S dopants can donate up to $10^7$ electrons/Te-NW (~1000× higher than intrinsic hole concentration), thus effectively changing the nature of the majority charge carriers. Grosse et al. observed a similar effect of a negative Seebeck coefficient in Te when thermally excited electron concentrations were comparable to hole concentrations $(6.1 \times 10^{14}/cm^3)$[55]. We can explore the effect of shifting $E_F$ on the Seebeck coefficient by modeling it as a function of the position of $E_F$ in the film (Supplementary Note 4 and Supplementary Figs. 23–26)[64–66]. In traditional TE materials, wherein the $E_F$ lies close to the band edge or deep within the band, a decrease in the charge carrier concentration would predict an increase in the Seebeck coefficient. However, in semiconductors such as Te where the $E_F$ lies well within the bandgap, decreasing the charge carrier concentration results in a decrease in the Seebeck coefficient due to a bipolar effect and subsequently leads to a change in its sign once the majority charge carrier inverts (Supplementary Note 4). This shift in the Seebeck coefficient for different doping concentrations can thus be thought of as a direct indicator of changes in the $E_F$.

While surface-doping solution-processed nanomaterials with small molecules have been previously demonstrated, stability and durability have historically impeded the technological implementation of such nanomaterials (for a detailed comparison of various reports, refer to Supplementary Note 5 and Supplementary Tables 1, 2). Talapin et al. and Wang et al. achieved transient n-type doping in thin films of p-type lead selenide nanocrystals with hydrazine[31,67]. However, hydrazine is a nonideal n-type dopant:

it is volatile, extremely toxic, and chemically unstable. As a result, the films lose their n-type character over time under vacuum or ambient pressures due to desorption of hydrazine. In addition, these nanocrystals are no longer solution-processable[68] and hence, incompatible with strategies for low-cost fabrication. In contrast, we use environmentally benign $S^{2-}$ salts to enable an effective doping strategy in water under ambient conditions resulting in doped NWs that, unlike hydrazine-doped materials, are stable for months and suitable for steady-state device studies. Although our thin films were cast in air, and the metal contacts for TE transport studies were evaporated post-deposition in high vacuum, our NW films retain their n-type behavior. To monitor the doping stability, we tracked the thermopower, under continuous sample operation, over long time periods. Figure 4c, d presents stability tests of samples over a period of 12 and 72 h, respectively; the thermopower range is clearly tuned by controlling the extent of $S^{2-}$ doping. Significantly, all doped NWs proved extremely robust, demonstrating <5% maximum standard deviation in the thermopower in all tested devices. To the best of our knowledge, this is the first reported study showing stable TE properties in doped nanomaterials over multiday continuous operation. Under inert conditions, these NWs are stable even for years. Figure 4 demonstrates the performance of one such set of NW films that were deposited nearly 2 years ago. Such stability is critical for integration of these materials into industrial applications.

Our covalent atomic resurfacing approach holds promise not only for TE applications but also lays a solid yet facile platform for a broad class of devices based on p-n homojunctions. While we have evidence for n-type doping via thermopower measurements (open-circuit), to demonstrate the generality of this approach for other classes of electronic devices such as diodes and transistors, we need to test our Te-NW films under bias. All of the observed trends in the Seebeck coefficient in the $S^{2-}$-doped Te NWs are in agreement with electrolyte-gated thin-film transistors measurements where electrical transport properties of the doped NW films were probed under varied electrochemical potentials (Fig. 5a and Supplementary Fig. 27)[69]. Figure 5c shows the transfer characteristics for transistors made from undoped Te NWs (corresponding output characteristics in Supplementary Fig. 27). As expected, the films start to conduct (drain current, $I_D$, increases sharply) when the interface potential ($V_G$) is biased negatively, which indicates that the film has positively charged holes available for conduction. More importantly, even at the highest positive bias, the films are not conductive, which indicates that only holes are involved in charge transport, no electrons are available for charge transport and hence $E_F$ lies close to the valence band edge (Fig. 5b). For intermediate doping (~1.5% sulfur atomic concentration) where Seebeck coefficients are nearly 0 µV/K, we note that both electron and hole conduction are achieved—a classic case of ambipolar transport—indicating that the $E_F$ lies close to the middle of the bandgap (Fig. 5c). In contrast, for the heavily doped Te-NW system (~2.5% sulfur atomic concentration), we observe hardly any conduction when the films are negatively biased whereas there is a sharp turn-on upon positive bias (Fig. 5d) with large electron conduction. This implies that the $E_F$ shifts closer to the conduction band edge—a signature of n-type doping corroborated by the WF results. Interestingly, while doped NWs show almost no hysteresis, the undoped ones show a large hysteresis implying the presence of long-lived trap states. Referring back to our XRD and XPS data (Supplementary Figs. 2 and 20), we observed $TeO_2$ peaks in the undoped NWs and none for the doped ones. This lends further credence to our assumption that $S^{2-}$ atoms might be passivating any existing dangling bonds on unbound surface Te atoms and thus prevent surface oxidation, and/or possibly filling any existing

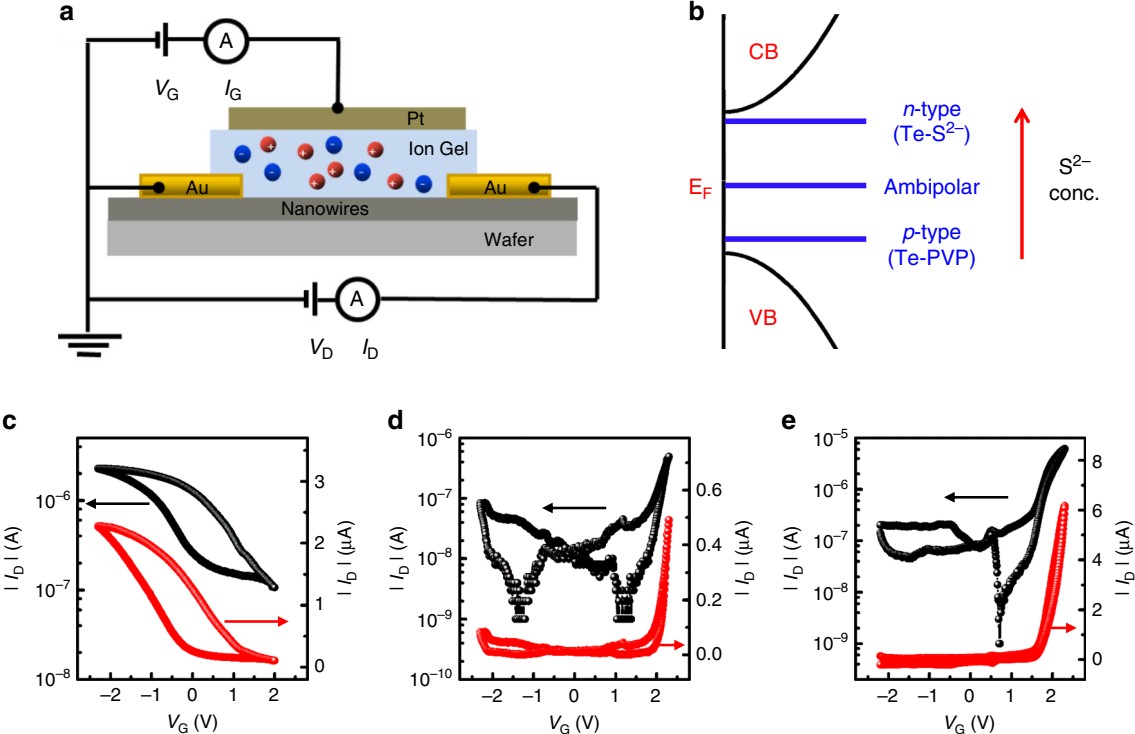

**Fig. 5 Fermi level shifts with doping. a** Schematic cross-section (not to scale) of ion-gel-gated thin-film transistors used to characterize the electrical properties of the doped NWs. The length and width of the channel were 100 μm and 2 mm, respectively. Red and blue circles represent positive and negative ions, respectively. **b** Energy level diagram depicting the relationship between the location of the Fermi energy ($E_F$), band-edges, doping concentration, and corresponding nature of charge carriers. CB and VB refer to the conduction band and the valence band of Te, respectively. Transfer characteristics for the **c** undoped Te-PVP NW sample (with $V_D = 0.1$ V) showing *p*-type transport, **d** intermediate-doped Te NWs (~1.5% atomic concentration of sulfur, with $V_D = -0.3$ V) showing ambipolar transport. **e** Heavily doped Te NWs (~2.4% atomic concentration of sulfur, with $V_D = -1.5$ V) showing *n*-type transport. Black and red curves plot the characteristics on logarithm and linear scales respectively.

mid-gap trap states. These electrical transport trends support our hypothesis that the change in sign of the Seebeck coefficient is attributed to a shift in the $E_F$ due to $S^{2-}$ doping.

Since we report this surprising *n*-type transport behavior in Te and there exists no prior report or analogue for *n*-type Te, we are unable to compare our results to others. We surmise that this is because no group has inverted carrier type in Te previously, further underscoring the power of this approach for modulating the electronic properties of even classical systems in unprecedented ways. In order to elucidate the doping mechanism and the charge-transfer mechanism in these atomically re-engineered and chemically resurfaced NWs along with the effect of the dopant band, we perform temperature-dependent electrical conductivity studies (Fig. 6). While the PVP-capped *p*-type Te NW films demonstrate activated (semiconductor-like) transport behavior as expected, surprisingly, the conductivity trends for the sulfur-doped *n*-type Te NW films point to metallic or band-like conductivity (i.e., conductivity decreases with increasing temperature) below 225 K and switch to an activated transport regime (i.e., exponential increase of conductivity with increasing temperature) above 225 K (Fig. 6c). Whilst conductivity values are only 0.5 S/cm at room temperature, they rise rapidly with increasing temperature, increasing by over two orders of magnitude and exceeding ~50 S/cm at ~150 °C, consistent with activated transport. These conductivity values thereby become comparable to high-performance TE materials at these elevated temperatures and have the potential rise even further at higher temperatures.

To understand the temperature-dependent results, we examine the band structure and DOS for both the doped and undoped

systems (Supplementary Note 2 and Fig. 3).The calculated DOS show the presence of a new dopant band close to the conduction band of Te in the sulfur-doped samples (Fig. 3e). Combining both results leads us to believe that below 225 K all the carriers are restricted to the dopant band with the Fermi level pinned inside the dopant band (hence the band-like transport), which would intuitively have extremely low mobility due to sparse DOS (hence the low conductivity). Similar Fermi-level pinning has been observed in Ti:PbTe resonant doping[52]. Above 225 K, carriers gain adequate thermal energy to hop into the conduction band (Fig. 6b) where they are free to conduct with extremely high mobilities, which leads to the sharp conductivity increase (Fig. 6c). We estimate the activation energy in this regime to be around 336 meV (Supplementary Figs. 28, 29). Referring back to our earlier predictions regarding the origin of the *n*-type conductivity, where we estimated that even at 2.5% concentration of dopant, the surface-bound sulfur dopants can provide 1000 times more electrons (assuming 100% carrier-doping efficiency) as compared with intrinsic hole concentrations ($4.8 \times 10^{17}$ carriers/cm$^3$). Thus, our estimations indicate carrier concentrations in the doped NWs ought to be nearly $4.8 \times 10^{20}$ electrons/cm$^3$. With this carrier concentration and an activation barrier energy of 336 meV (obtained from the temperature-dependent conductivity studies), we estimate the concentration of free carriers in the conduction band at 300 K to be nearly $7 \times 10^{14}$ electrons/cm$^3$. Assuming bulk electron mobilities (obtained from hall measurements at high temperatures due to promotion of electrons from valence to conduction band) of 2380 cm$^2$ V$^{-1}$ s$^{-1}$ [55], we estimate the conductivity at 300 K to be nearly 0.3 S/cm, respectively. Experimental value of conductivity for the same sample at 299

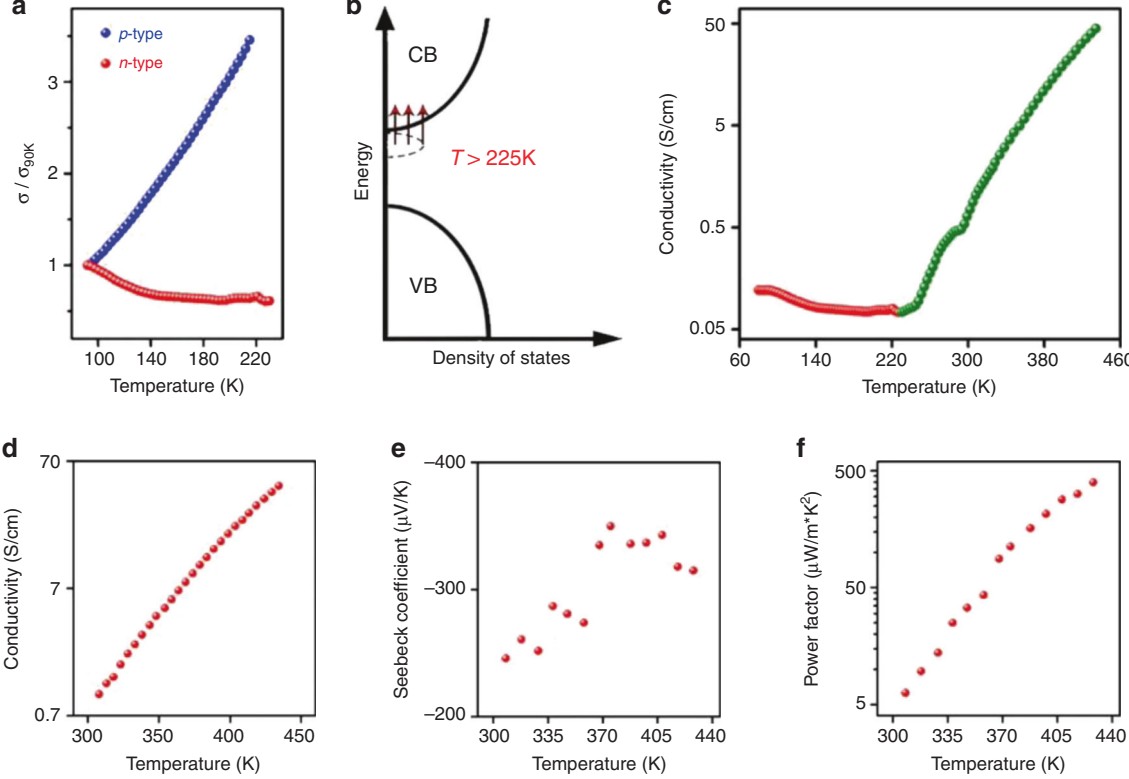

**Fig. 6 Thermoelectric transport properties in surface re-engineered tellurium nanowires. a** Temperature-dependent conductivity ($\sigma$) measurements (from 90 to 225 K) on thin films of undoped *p*-type (Te-PVP, blue spheres) and sulfur-doped *n*-type (Te–S$^{2-}$, red spheres) tellurium nanowires normalized to the respective conductivity values at 90 K ($\sigma_{90K}$). While the undoped nanowire films show activated (semiconductor-like) transport with conductivities increasing with increasing temperature, the doped nanowire films demonstrate band-like transport with decreasing values of conductivities with increasing temperature. **b** Proposed doping scheme wherein sulfur dopes tellurium *n*-type and most of the electrons are located in the dopant band with the Fermi level inside the band (similar to a metal) at low temperatures (below 225 K), and hence the band-like transport behavior. Above 225 K, the electrons gain sufficient thermal energy to get promoted to the conduction band. **c** Temperature-dependent conductivity measurements for the *n*-type (Te–S$^{2-}$) samples wherein above 225 K the conductivity increases sharply with increasing temperature. Temperature-dependent **d** electrical conductivity, **e** Seebeck coefficient, and **f** power factor of the *n*-type Te nanowire film demonstrating extremely high monotonously increasing power factors with increasing temperature.

| Table 1 Generalizability of doping approach. | | | | | | |
|---|---|---|---|---|---|---|
| **Dopant** | **PVP (Undoped)** | **Na$_2$S** | **NaHS** | **(NH$_4$)$_2$S** | **K$_2$S** | **KHS** |
| $S$ ($\mu$V/K) | 524 ($\pm$25) | −307 ($\pm$10) | −242 ($\pm$15) | −183 ($\pm$19) | −232 ($\pm$5) | −197 ($\pm$13) |
| $\sigma$ (S/m) | 1.45 ($\pm$0.18) | 1.25 ($\pm$0.14) | 1.19 ($\pm$0.15) | 1.38 ($\pm$0.11) | 1.09 ($\pm$0.2) | 0.8 ($\pm$0.06) |
| Seebeck coefficient (S) and electrical conductivity ($\sigma$) of ~10-nm diameter Te NW films showcasing the generalizability of the *n*-doping approach with a broad range of benign sulfur-based dopants. | | | | | | |

K is 0.6 S/cm which is consistent with the physical picture that the electrical conductivity in our *n*-type systems originate from the charge carriers that are promoted into the conduction band from the dopant band.

Using the calculated DOS to estimate the Seebeck coefficients (details in Supplementary Note 4 and Supplementary Fig. 17), we observe a significant shift in the Seebeck coefficient toward negative values at a constant Fermi level when we compare the doped and undoped samples owing to the local band restructuring. With moderately high electrical conductivities but extremely high Seebeck coefficients, these materials seem suited to operate best at intermediately high temperatures. They demonstrate extremely high power factors (~500 $\mu$W m$^{-1}$ K$^{-2}$) at ~150 °C, almost an order of magnitude higher than state-of-the-art solution-processed PDI systems[10]. As emphasized earlier, the purpose of the work is to investigate a unique doping strategy in

nanomaterials with an eye toward improving TE properties. Not only have we succeeded in demonstrating a remarkably simple, tunable, and general yet robust doping scheme, we also obtain materials with extremely promising TE properties with power factors approaching ~500 $\mu$W m$^{-1}$ K$^{-2}$, which are comparable with commercial solution-synthesized and solution-processed TEs. We believe that we have not yet reached the boundaries of the material properties and there is a scope of obtaining even higher *ZT* values, but again the emphasis of this report is on the methodology for transforming the electronic properties of materials. This is but one example of the potency of our methods.

To showcase the versatility of our technique, we used several different dopant molecules with S$^{2-}$ and SH$^-$ moieties under the same conditions and in each and every case, we obtain *n*-type behavior (Table 1). The design rules that we present for doping are extremely general and can be broadly adopted across many

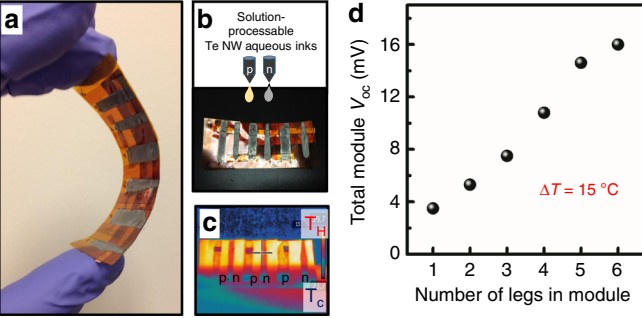

**Fig. 7 Thermoelectric generator demonstrations for in-plane devices. a** A flexible module prepared from solution-processable Te NW inks in an unconventional thin-film leg-geometry. **b** Module n- and p-type legs drop-cast onto Kapton substrate to make the array shown in **a**. **c** Infrared image depicting a temperature gradient of 15 °C established across the thin-film module. **d** Total module open-circuit voltage ($V_{OC}$) generated as a result of the Seebeck effect as a function of the number of n- and p-type legs connected in series in the module depicted in **a–c**. Additive nature of the $V_{OC}$ confirms the proper working of the module.

classes of semiconducting and metallic materials as well. Our preliminary experiments demonstrate n-type switching behavior in bismuth telluride NWs as well. This approach, thus, provides us with a technique that radically alters how we judge attractive material candidates for applications, like TEs. More generally, this approach of doping and band modification can now be applied into other established TE systems like PbTe, where only one type of transport is favorable and if one can render the other band also favorable for TE transport, one could potentially make significant advancements in the field. In addition, our doping methodology abstracts the dopants from the transport channel—so without any intrinsic carrier–carrier and carrier–vacancy limitations, it actually (in principle) enables one to achieve much higher doping levels without introducing ionized impurity scattering thus pushing the electrical boundaries of existing materials even further. This powerful methodology will help expand the library of attractive and viable materials, push the tunability of their electronic properties and help discover alternatives to break the reliance on rare materials currently in use. In light of TE applications, we tackle two important issues here. Materials in which both bands could be favorable for TE transport but are not used because of an inability to dope it one way or the other can now be doped with this approach. In materials, which could be doped both n- and p-type but one band was unsuitable for TE transport, one could now envisage increasing the local DOS around the "unfavorable" band by the dopant band and modifying the TE transfer parameters in ways that could make transport through this band appealing as well.

Typically, most current TE modules consist of a p-type and an n-type material based either on bulk crystals or SPS nanostructures, micro-machined or diced into rigid millimeter-thick pillar structures, which are then serially assembled with metal interconnects. However, if the materials have substantially different conductivities (thermal or electrical), the legs need to be fabricated with different cross sections, which considerably increases the complexity of module fabrication. In addition, if there is a mismatch between the thermal expansion coefficients of the two TE materials, strain will develop in the module, leading to a degradation of its structural integrity. During the dicing and assembly process, a significant fraction of material is wasted, and rigid module designs also restrict deployment to somewhat limited applications where heat flux can be captured from flat surfaces with regular geometries. As emphasized earlier, colloidal

NWs provide the unique advantage of solution processability that can be utilized for cheap patterning on both rigid modules and flexible substrates as well as introducing the possibility to realize more complex TE geometries or thermal energy harvesting fabrics that are challenging to fabricate with conventional bulk semiconductor TEs[39]. We harnessed these unique advantages to prepare TEG modules in an unconventional thin-film leg-geometry on a flexible Kapton substrate (Fig. 7a–c) simply by drop-casting our doped and undoped Te NWs onto the substrates in ambient conditions. Figure 7d shows the performance of this "monomaterial" device wherein the six legs (three p-type and three n-type) are connected electrically in series and thermally in parallel. Under a temperature gradient of nearly 15 °C (Fig. 7c), the additive nature of the open-circuit voltage proves that the device functions as one would expect from a conventional TEG. In addition, our n- and p- legs have thermopower values that are of equal magnitude and opposite signs and show great self-consistency among the three legs, respectively.

## Discussion

In this manuscript, we reveal chemically mediated interfacial doping as an underexplored, yet powerful doping paradigm for nanostructured TEs. Traditional doping schemes use electronic doping by utilizing the difference in valence states between dopant and host atoms while charge-transfer doping achieves the same by using band offsets between semiconductors. We apply the principle of resonant DOS band engineering traditionally used to augment Seebeck coefficients in bulk semiconductors and demonstrate complete inversion of the majority carrier type in a conventional p-type semiconductor nanostructure. Our report combines isoelectronic, charge transfer, and resonant doping in a single system to demonstrate n-type transport in Te. Our findings show that precision chemical doping of nanostructures enables intricate control over the carrier transport of the films, generating extremely stable and robust p-type, n-type, and ambipolar characteristics from the same material system. These findings further suggest that similar means could induce p-type doping in n-type materials. With this methodology, we tune the Seebeck coefficient of Te over nearly three orders of magnitude and generate extremely high n-type power factors, enabling facile construction of flexible TE modules with both p- and n-legs comprised the same solution-processed material without any post-processing steps. The combination of extreme carrier type tuning while leaving the host lattice intact is a potent electronic dial with strong implications for thin-film TEs, light emitting devices, and electronic logic.

## Methods

**Chemicals and substrates**. Te dioxide (99.9995%), sodium hydroxide (ACS reagent, ≥97.0%, pellets), PVP, average molecular weight (~55,000), ethylene glycol (ReagentPlus®, ≥99%), hydrazine hydrate (78–82%, iodometric), platinum wire (99.9%), ammonium sulfide solution (40–48 wt% in $H_2O$), and methanol (reagent grade, 98%) were purchased from Sigma Aldrich. Acetone (J. T. Baker®), dichloromethane (BDH®), and isopropyl alcohol (ACS Grade) were purchased from VWR International. Sodium sulfide (anhydrous, min. 99.5%), sodium sulfide nonahydrate (ACS reagent, >95%), sodium hydrosulfide hydrate, potassium sulfide (anhydrous, min. 95%), and potassium hydrosulfide (anhydrous, min. 95%) were purchased from Strem Chemicals. 1-ethyl-3-methylimidazolium bis(tri-fluoromethylsulfonyl)imide ([EMIM][TFSI]) was purchased from Solvent Innovation GmbH (Germany). All chemicals were used as delivered without further purification.

Glass substrates, 9.5 × 9.5 mm, and 0.2 mm thick, were purchased from Thin Film Devices. <100>-oriented, boron-doped Si wafers (resistivity = 0.005–0.01 Ω cm, thickness = 525 ± 25 μm) coated with 300 nm of thermal oxide ($SiO_2$) were purchased from Silicon Valley Microelectronics.

**Sample characterization**. XRD, SEM, energy-dispersive X-ray spectroscopy (EDS), transmission electron microscopy (TEM), TGA, XPS and fourier transform

infrared spectroscopy were used to characterize the size, shape, structure, composition, and optical properties of the NWs.

For wide-angle XRD, a Bruker-GADDS 8 microdiffractometer was utilized to collect wide-angle powder patterns (Cu-Kα). Samples were prepared from concentrated dispersions of Te NWs in water. Films of these nanocrystals were deposited onto heavily doped Si wafers covered with a thermally grown 300-nm-thick $SiO_2$ layer.

For TEM, a Libra120 and an FEI Technai G20 Super Twin Lab6 microscope were used to image the NWs with acceleration voltages of 120 and 200 kV, respectively. Each sample was prepared by depositing a drop of dilute NW dispersion in methanol onto a 400-mesh carbon-coated copper grid and allowing the solvent to evaporate at room temperature.

The HAADF-STEM and EDX maps were performed on a FEI TitanX 60-300 microscope equipped with Bruker windowless EDX detector at an acceleration voltage of 60 kV. The sample was prepared by depositing a drop of dilute NW dispersion in water onto a lacy carbon 400-mesh copper grid placed upon a hot plate covered by clean filter paper. The film then was gently heated to 60 °C to allow the solvent to evaporate. The STEM-EDX and EDX line scans were used to analyze the elemental distribution in S-doped Te NWs. The images clearly show that sulfur is concentrated in the surface region, while Te is concentrated in the core region.

SEM images were recorded and EDS analyses were performed on a Zeiss Gemini Ultra-55 Analytical Scanning Electron Microscope using beam energy of 5 kV and an In-Lens detector.

TGA were acquired using a Thermal Advantage Q20 calorimeter at a scan rate of 10 °C min$^{-1}$. An indium standard was used to calibrate the instrument, and nitrogen was used as the purge gas.

XPS was performed using a monochromatized Al Kα source ($h\nu = 1486.6$ eV), operated at 225 W, on a Kratos Axis Ultra DLD system at a takeoff angle of 0° relative to the surface normal and a pass energy for narrow scan spectra of 20 eV, corresponding to an instrument resolution of ~600 meV. Survey spectra were collected with pass energy of 80 eV. Spectral fitting was performed using Casa XPS analysis software. Spectral positions were corrected by shifting the primary C 1s core level position to 284.8 eV, and curves were fit with quasi-Voigt lines following Shirley background subtraction. Samples were deposited on conductive Si substrates.

UPS was performed on a Thermo K-Alpha Plus instrument using a He I source (21.2 eV) and a −5 V applied bias. Values for the WF for the Te-PVP and Te–S$^{2-}$ NW film were extracted via UPS spectra. All thin-film surfaces were cleaned using an Ar cluster gun (6000 eV/cluster, 150 atoms/cluster, 15 s) before all UPS experiments. WFs were extracted as the difference between the UPS radiation source energy and the secondary electron cutoff energy (SECO) using linear fits, whereas ionization energies were calculated by measuring the valence band onset. SECO was determined by fitting the onset curve and finding the point of intersection of that fit line. All data were corrected with respect to the applied bias and the fermi onset energy of a gold standard.

**Ion gels**. A symmetric poly(styrene-block-(ethylene oxide)-block-styrene) (PS-PEO-PS) triblock copolymer was synthesized, as previously described[70]. To prepare the ion gels, PS-PEO-PS and 1-ethyl-3-methylimidazolium bis(trifluoromethylsulfonyl)imide [EMIM][TFSI] (1:9 by weight) were dissolved in dichloromethane ($CH_2Cl_2$). The solution was stirred overnight, and then poured into a Petri dish. Dichloromethane was slowly evaporated at room temperature for 24 h, and the ion-gel solution was further dried under vacuum for an additional 24 h. After complete evaporation of the solvent, transparent ion gels were formed. They were stored in a nitrogen glove box until needed.

**Synthesis and doping of nanowires**. Te-NWs were synthesized by established literature protocols[43,44]. After numerous rounds of purification with water and methanol (typically 4–5), the resulting NWs were dispersed in water and were stored in ambient conditions until needed. For the surface doping with S$^{2-}$, a typical reaction involved adding requisite amounts of $Na_2S$ to a 4–5 mg/ml dispersion of Te NWs in water in a glass jar with continuous stirring overnight (~16 h). After the reaction was stopped, the doped NWs were centrifuged for ~45 min. While the NWs crashed out of the dispersion, any unreacted $Na_2S$ and PVP stripped from the NWs remained in the supernatant, which was discarded. The precipitated NWs were redispersed in water and isolated again with centrifugation. The process was repeated at least four times to ensure the removal of any unbound species and obtain a clean product. Finally, the NWs were resuspended in ultrapure 18 MΩ water and stored for further use.

**Thermoelectric measurements**. For the TE, XRD, XPS measurements and SEM imaging, we used quartz substrates that were sonicated in acetone, isopropanol, and methanol (10 min in each), rinsed with methanol, dried at 100 °C for nearly 15 min and then UV ozone treated for 30 min. Films of NWs (typically 5 microns thick) were then drop-cast from a 100 mg/ml dispersion in water and heated at 70 °C for nearly 30 min. Contacts (100 nm Au) were patterned on the film using evaporation. Electrical conductivity was measured using a Keithley 2400 Sourcemeter in 4-wire

Van-der-Pauw configuration. Thermopower (Seebeck coefficient) was measured using a homemade setup consisting of two Peltier devices (Ferrotec) spaced ~4 mm apart. Current was driven through the devices in opposing polarities, resulting in a temperature gradient about room temperature which varied with the magnitude of the current. The temperature of the sample was measured using two T-type thermocouples mounted in micromanipulators. Thermal contact was ensured by utilizing Si thermal paste (Wakefield Thermal S3 Solutions). Typically, five different gradients were employed, with ten voltage samples taken and averaged using an Agilent 34401 multimeter with an equilibration time of 2–3 min between temperature changes. Data for both conductivity and thermopower were acquired using homemade Labview programs. Uncertainty in all measurements is either shown directly in the reported data, the captions, or described in the "Methods" section or the Supplementary Information. The Seebeck coefficient data has measurement uncertainties of roughly 2–3% and hence are captured within the data points unless otherwise noted. Errors in conductivity (typically in the order of 5–10%) are shown in Supplementary Information and not in the main text for clarity.

**Thin-film transistor measurements**. For the thin-film transistors, we used Si/$SiO_2$ wafers which were sonicated in acetone, isopropanol, and methanol (10 min in each), rinsed with methanol and dried at 100 °C for nearly 15 min. Films of NWs (typically 100–200 nm thick) were then either drop-cast from a 5 mg/ml dispersion in water and heated at 70 °C for nearly 30 min, or spin-coated (20 s at 1700 r.p.m. followed by 30 s at 3000 r.p.m.) from a 50 mg/ml dispersion in methanol and were dried at room temperature for nearly 1 h. Source and drain contacts (50 nm Au) were patterned on the film using evaporation. For the gate dielectric, an ion gel comprising of 10 wt% of the triblock copolymer poly(styrene-block-ethylene oxide-block-styrene) and 90 wt% of the ionic liquid 1-ethyl-3-methylimidazolium bis (trifluoromethylsulfonyl)imide was spread over the active channel. Finally, a Pt foil was placed over the ion gel to serve as the gate electrode. Current–voltage ($I$–$V$) characteristics were measured using an Agilent 4155C semiconductor parameter analyzer.

**Density functional theory calculations**. The Vienna ab initio software package[60] was used to perform the DFT calculations, along with the projector augmented-wave method[61] to describe the ion–electron interactions and the generalized gradient approximation[62] within the Perdew–Burke–Ernzerhof framework[63] as the exchange-correlation functional. The surface calculation of Tewas modeled by a slab supercell with a vacuum thickness of 20 Å to separate the slabs and a Te slab thickness of >17 Å to prevent interaction between two termination surfaces. The lattice parameters of the supercell were fixed at its bulk value in the slab in-plane direction, and all the atoms were relaxed freely until the forces are <0.01 eV/Å. All calculations used a plane-wave cutoff of 500 eV and $21 \times 21 \times 1$ mesh for k-space to obtain converged results. Moreover, the magnetic moment was not considered in all the calculations as Te and sulfur are not spin-polarized. In the physical adsorption case, the S adatom is loosely attached to the Te surface by weak Van-der-Waals forces, whereas in the case of chemical adsorption, stronger bonding exists between the two atoms resulting in a shorter bond length. In the calculation, firstly, the structures were fully relaxed to reach their equilibrium configurations. Then, the charge densities on the bare Te surface and the isolated S adatom were calculated separately by using the same atomistic position from the adsorbed structure but fixing each atom individually in its position to prevent the relaxation. By doing so, it was possible to compare the charge densities of sulfur, Te, and sulfur on Te, respectively and extract the charge-transfer effect between sulfur and Te.

**Thermoelectric modules**. For TE module demonstration, doped and undoped Te NW dispersions were briefly agitated prior to deposition. Individual p- and n-type legs were drop-cast onto Kapton substrates and dried at 90 °C on a hot plate. Electrical contacts were made using silver paint (PELCO, Ted Pella) or thermally evaporated gold. Kapton tape was pre-cleaned with soap and water and exposed to 5 min of UV/ozone treatment prior to NW dispersion deposition. To establish a temperature gradient across the device, one side of the TE module legs was placed in contact with a hot plate/heating block, while the other side of the legs in the TE module was exposed to ambient air. The temperature gradient was quantified using an infrared imaging camera (FLIR OSXL-E50). To establish module performance, electrical contact with the appropriate number of p/n legs in series was made using thin tip movable probes; the $V_{oc}$ generated by the TE module under a temperature difference was measured using an Agilent voltmeter. All measurements were carried out under ambient conditions in air.

## Data availability
The datasets generated during and/or analyzed during the current study are available from the corresponding author on reasonable request.

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

## Acknowledgements

This work was partially performed at the Molecular Foundry, Lawrence Berkeley National Laboratory, and was supported by the Office of Science, Office of Basic Energy Sciences, Scientific User Facilities Division, of the U.S. Department of Energy under Contract No. DE-AC02-05CH11231. The computational work was performed under the infrastructure of the Materials Project which is supported by Department of Energy's Basic Energy Sciences program under Grant No. EDCBEE. The TEM characterizations were performed at the Molecular Foundry at LBNL, supported by the Office of Science, Office of Basic Energy Sciences, of the U.S. Department of Energy under contract No. DE-AC02-05CH11231. B. R. gratefully acknowledges the Department of Defense, AFOSR, for fellowship support under the National Defense Science and Engineering Graduate Fellowship (DOD-NDSEG), 32 CFR 168a under contract FA9550-11-C-0028. E.W.Z. and M.P.G. gratefully acknowledge the National Science Foundation for fellowship support under the National Science Foundation Graduate Research Fellowship Program. The authors would like to thank Raffaella Buonsanti and Chris Dames for insightful discussions and thoughtful feedback, as well as Hilda Buss for synthesis of the block copolymer and Qintian Zhou and the staff of LBNL, particularly A. Brand, T. Mattox, and T. Kuykendall, for their support.

## Author contributions

A.S. and J.J.U. conceived and designed the experiments. A.S. performed the synthesis, doping, structural and thermogravimetric analyses. B.R., J.D.F. and A.S. performed electron microscopy. A.S. and N.E.C. performed thermoelectric and electric characterization. E.W.Z. X-ray photoelectron spectroscopy and ultraviolet photoelectron spectroscopy. M.L. and K.A.P. performed density functional theory calculations. F.Y. modeled the thermoelectric properties. B.R. designed the thermoelectric module. M.P.G. synthesized materials for the scanning transmission electron microscopy energy-dispersive X-ray spectroscopy performed by Y.-Q.Z. A.S., N.E.C., B.R., R.A.S., and J.J.U. compiled and analyzed the data and wrote the manuscript. All authors discussed the results and commented on the manuscript.

## Competing interests

A.S., B.R., M.L., F.Y., J.D.F., N.E.C., R.A.S., and J.J.U. are listed as inventors on a patent filed by the University of California, Berkeley in September 2016, entitled "Surface Doping of Nanostructures" for the surface doping procedure described in Fig. 1.
