## [Peer Review File · Nature Communications]

Reviewers' Comments:

Reviewer #1:

Remarks to the Author:

The authors present an interesting approach to doping Te nanowires to convert them from p-type to n-type with relatively high thermoelectric performance. The major claim of the manuscript is that the addition of S²⁻ to Te NWs results in a surface doping by the S²⁻ and the introduction of resonant levels in the DOS at the conduction band edge. The explanation is supported by a combination of experimental techniques combined with density functional theory calculations. Despite the experimental and theoretical data presented, there are many questions that remain and the arguments presented are not strongly supported. Without strong support for the proposed mechanism, the manuscript should not be published in Nature Communications. The thermoelectric performance alone is also not sufficient for publication, as other solution processed inorganic thermoelectric materials have shown high performance and in this manuscript the thermal conductivity is not measured and ZT cannot be determined. The below comments indicate specific issues that should be further addressed if the manuscript is to be reconsidered by Nature Communications.

1. The authors are not adequately accounting for all that is occurring upon S²⁻ doping that could be influencing the material properties. A large amount of TeO₂ is apparent in the samples prior to S²⁻ doping, as shown in both XPS and XRD data, but after doping the TeO₂ signature is completely eliminated from the XRD data and the TeO₂ peaks in the XPS data are reduced. TeO₂ is a wide bandgap semiconductor and is p-type. If this TeO₂ is more localized at the NW surfaces, which seems reasonable, it should be influencing transport between nanowires. Elimination of part or all of this high bandgap TeO₂ alone may have a huge influence on electronic properties.

2. Building on point 1, the role of interfaces and transport between nanowires is largely neglected. The elimination/reduction of TeO₂ and PVP at the NW surfaces, as well as the highly crystalline surfaces, obtained with S²⁻ doping will improve electronic contacts between NWs. The electronic effects at these NW-NW contacts are not adequately addressed, and are expected to be significant.

3. The authors propose that the S atoms remain near the surface, which is a major selling point of their doping strategy, but there is no evidence supporting where the S atoms are located. EDS line scans, EELS imaging, etc. is necessary to show whether the S atoms actually remain surface localized.

4. The presence of a large amount of S in the undoped samples is concerning. The S peak in the XPS sample for the undoped Te NWs is much more intense than the peak from the S²⁻ in the doped samples. It would be helpful if the XPS data was used to calculate atomic concentrations to provide a gauge on the amount of S present in all samples.

5. Measurements of charge carrier densities in these materials, such as through Hall effect measurements, are necessary for understanding the mechanism of S²⁻ treatment. Carrier densities are discussed, but no direct measurement of these carrier densities are presented. Measurements of carrier densities are particularly important given that S²⁻ doping will simultaneously influence both the electronic contact between NWs and the carrier densities.

Smaller points:

In Figure S1 there is not much contrast between the crystalline Te and the amorphous polymer. C and Te should show a large amount of contrast and it is possible that this amorphous polymer layer is actually TeO₂.

Ultraviolet photoelectron spectroscopy measurements to compare to calculated band structures and to directly probe the difference between the Fermi energy and the valence band onset in the different samples would be helpful.

The temperature dependent conductivity measurements of the doped samples are interesting, particularly the change in activation energy occurring after 225 K. Seebeck measurements over this entire range would be helpful in interpreting what is occurring, but Seebeck measurements are only presented with temperatures above 225K.

How were the WFs and IEs measured?

Assuming mobilities of $2380 \text{ cm}^2/\text{Vs}$ based on bulk Te, or perhaps single crystal Te (I could not access the reference to find out), is not a fair assumption. NWs, and particularly NW films, display much lower mobilities.

More credit should be given to other solution processed thermoelectric materials. Currently, the manuscript focuses on comparison to organics, but many inorganic materials are also able to be solution processed. For example, Nature Energy volume 3, pages301–309 (2018) and Scientific Reports volume6, Article number: 33135 (2016).

The figures in the SI should be numbered in order of when they are mentioned in the text.

Initial XRD should be shown for the undoped Te NWs, as the XRD data in Figure S2 is after weeks of exposure to ambient atmosphere.

Reviewer #2:

Remarks to the Author:

In this paper, the authors report a surface-engineered doping strategy for tellurium nanowire by sulfur anion ligands in which p-type properties were dramatically changed to n-type. To verify the effectiveness of this approach, the authors systematically analysed structural and electrical changes of tellurium nanowire as increasing doping concentration of sulfur. Their fundamental study is well organized, and the following results are also very impressive. Furthermore, the authors successfully produced thin film transistors to demonstrate its generality over other electronic devices, and flexible thermoelectric generator to show its excellent applicability. Accordingly, I recommend the publication of this manuscript in Nature Communications since the subject matter can potentially appeal to a broad audience in the thermoelectric society. Some minor comments are as follows.

1. What is the possible range of doping? The authors need to report on the doping limits of sulfur for tellurium.
2. I wonder if the authors have tried to use TE nanowire with different diameter? Since this strategy is strongly related to surface chemistry, the surface to volume ratio of Te nanowires could influence to doping effect.
3. Identifying the doping possibility for other isovalent elements may help clarify the doping mechanisms suggested in this manuscript. (i.e. Sulfur doping for selenium or selenium doping for tellurium.)

Reviewer #3:

Remarks to the Author:

This work reports on the modulation of electrical transport and thermoelectric properties in Te nanowires by interfacial resurfacing/doping with S. According to the authors, there are mainly two effects of S-doping: one is the shift of Fermi level, and the other is the formation of a new, dopant band. In this manuscript, to my feelings, the two effects are somewhat mixed up. It is not clear whether the doping effects are body effects (like conventional effects in bulks) or interface effects. The key phenomenon of p-n conversion is not clearly understood. It is reasonable that after

doping, S is partially negative and Te is positive, but this cannot explain why the system as a whole is n-type. Also, the resonant dopant band lacks substantiation. In addition, the device efficiency is not discussed, so the quality of the material and the device cannot be evaluated. In addition, some concerns or comments on the technical details are also listed below.

1. As shown in Fig. 2 (d) and (e), there is little difference in band structure between undoped and S-doped Te except for the "dopant band" in the latter. Does the dashed line represent the Fermi level? If so, undoped Te (Fig. 2d) should also be n-type.
2. The evidence for the resonant band just from calculation is inadequate. Perhaps experimental Pisarenko relation (Seebeck coefficient vs carrier concentration) or optical measurements can be helpful.
3. HR-TEM and/or other analyses are needed to show the actual position of S in Te matrix, i.e., substitutional, interstitial or precipitation, which is the basis to understand the change in electrical properties.
4. Before discussing S-doped Te, the authors should make it clear why intrinsic Te is p-type. What are the intrinsic defects? Then how are the defects affected by S doping?
5. Table 1: Are WF and IE obtained experimental or calculated values? Also, I suggest giving a schematic drawing to show WF and IE with respect to VBM, CBM and EF.
6. Fig. 4 and relevant analyses are quite confusing and difficult to understand. For example, the definition positive or negative biased VG is not clear. Line 387, page 11, "more importantly, even at the highest positive bias the films are not conductive..."? What does it mean? The former sentence says the film is p-type conductive. Please clearly illustrate this figure and consider moving it to SI since it is only supplemental to the p-n transition.
7. The text seems too lengthy with too many subjective or exaggerated expressions. I suggest authors carefully refining the words.
8. More references on Te as thermoelectric material should be cited such as Nature Communications volume 7, Article number: 10287 (2016).

Reviewers' comments:

Reviewer #1 (Remarks to the Author):

The authors present an interesting approach to doping Te nanowires to convert them from p-type to n-type with relatively high thermoelectric performance. The major claim of the manuscript is that the addition of S²⁻ to Te NWs results in a surface doping by the S²⁻ and the introduction of resonant levels in the DOS at the conduction band edge. The explanation is supported by a combination of experimental techniques combined with density functional theory calculations. Despite the experimental and theoretical data presented, there are many questions that remain and the arguments presented are not strongly supported. Without strong support for the proposed mechanism, the manuscript should not be published in Nature Communications. The thermoelectric performance alone is also not sufficient for publication, as other solution processed inorganic thermoelectric materials have shown high performance and in this manuscript the thermal conductivity is not measured and ZT cannot be determined. The below comments indicate specific issues that should be further addressed if the manuscript is to be reconsidered by Nature Communications.

We appreciate the Referee's comments and their positive feedback. We have used the Referee's remarks to further revise the manuscript. **Below we give a point-by-point response to the comments and indicate where revisions have been made (highlighted in yellow).**

Despite the experimental and theoretical data presented, there are many questions that remain and the arguments presented are not strongly supported. Without strong support for the proposed mechanism, the manuscript should not be published in Nature Communications. The thermoelectric performance alone is also not sufficient for publication, as other solution processed inorganic thermoelectric materials have shown high performance and in this manuscript the thermal conductivity is not measured and ZT cannot be determined.

We respectfully disagree with the Reviewer's comments that we do not provide a strong support for the proposed mechanism. We propose a **comprehensive experimental and theoretical framework** using a multitude of complementary techniques that unequivocally proves the doping mechanism. Rather than emphasizing our impressive thermoelectric performance for an all solution-synthesized and solution-processed nanomaterial, we believe our robust description and discovery of the doping mechanism that uses resonant band engineering to invert the carrier type in degenerately doped semiconductors is the primary focus of this work. Additionally, to the best of our knowledge, there exists no other stable, solution-synthesized and solution-processed n-type materials system that demonstrates high thermoelectric power factors of 500 $\mu\text{W}/\text{m}\cdot\text{K}^2$ without any hot-pressing and/or spark plasma sintering (which would render it unusable for use in flexible substrates and niche applications that require flexible form factors). We agree that while isolated observations will not prove a convincing case for the resonant band engineering claims and could be serendipitous, each and every result obtained by us are complementary to one another and supplement each other to provide a compelling case for the doping scheme. For the Reviewer's convenience, we summarize the observations below that substantiate our claims of the doping mechanism:

- 1) Sulfur dopants are not randomly distributed throughout the Te-lattice but are localized only on the surface.
- 2) DFT calculations showing the appearance of a new dopant band due to these localized Sulfur-states on the surface of the Te-nanowires
- 3) DFT calculations and XPS data demonstrating that electrons are localized in the S-states (and hence in the dopant band)
- 4) Work function (experimental) measurements demonstrating the reduction in work-function by 0.3 eV thus indicating a shift of the Fermi level towards the conduction band. The band gap of Te is 0.35 eV which implies that the work-function lies either in the dopant band or pretty close to the conduction band of Te.
- 5) Transistor measurements that show a shift in the Fermi level gradually from close to the valence band to the conduction band as a function of increased dopant addition – corroborating the work function shift
- 6) Seebeck coefficient change with S-addition – gradually from positive values to negative values through experimental results – again corroborating to the calculated Seebeck coefficient values from DFT calculations combined with modeling using Boltzmann Transport equations.
- 7) Temperature dependent electrical conductivity data – that demonstrates a switch in transport mechanism from localized transport within the narrow dopant band (dominated by scattering) to transport within the conduction band of tellurium (activated transport). The switch happened nominally at 220 K (~ 19-20 meV) that matches up qualitatively with the location of the shallow dopant band right below the Te-conduction band.

We review all these observations in the next few sections:

1) Sulfur dopants are not randomly distributed throughout the Te-lattice but are localized only on the surface: This claim of surface re-engineering is proven by new STEM/EDX results shown below:

Figure 2. Structural characterization of dopant location. **a**, high-angle annular dark-field scanning transmission electron microscopy (HAADF-STEM) and **b, c**, energy-dispersive X-ray (EDX) maps of sulfur and tellurium respectively in a sulfur-doped tellurium nanowire showing the presence of a minute amounts of sulfur in the sample **d**, line scans demonstrating that sulfur atoms are primarily concentrated on the surface of the nanowire.

The high-angle annular dark-field scanning transmission electron microscopy (HAADF-STEM) and energy-dispersive X-ray (EDX) maps were performed on a FEI TitanX 60-300 microscope equipped with Bruker windowless EDX detector at an acceleration voltage of 60 kV. The sample was prepared by depositing a drop of dilute nanowire dispersion in water onto a lacy carbon 400-mesh copper grid placed upon a hot plate covered by clean filter paper. The film then was gently heated to 60 °C to allow the solvent to evaporate. The STEM-EDX and EDX line scans were used to analyze the elemental distribution in S doped Te nanowires. The images clearly show that sulfur is concentrated in the surface region, while Te is concentrated in the core region.

Figure S4. STEM EDX spectrum before and after deconvolution: The spectrum shows the presence of a tiny amount of Sulfur in the S-doped Te samples. The approximate value from the spectrum is S: 0.38 at.%, Te: 99.62 at.%. However, since the S dopant concentration is too low, and the EDX data generates a systematic error around 1-2% for quantification considering the errors in background subtraction, data fitting etc., thus the absolute quantification value for S and Te are not accurate.

2) DFT calculations showing the appearance of a new dopant band due to these localized Sulfur-states on the surface of the Te-nanowires:

Figure 3. Density functional theory calculations showing charge transfer and evolution of a resonant dopant band with sulfur doping. d, e, Calculated surface density of states using density functional theory (DFT) for tellurium (*p*-type) and sulfur-doped tellurium (*n*-type) with CB and VB referring to the conduction and valence bands of bulk tellurium respectively. The dotted black line denotes the edge of the CB. A new dopant band (shaded ellipse) emerges close to the conduction band edge for the sulfur-doped tellurium.

3) DFT calculations and XPS data demonstrating that electrons are localized in the S-states (and hence in the dopant band)

Figure 3. Density functional theory calculations showing charge transfer and evolution of a resonant dopant band with sulfur doping. **a**, Illustration of the atomistic structure and (010) surface, of the hexagonal tellurium nanowire. The supercell configuration and the charge transfer effect for **b**, physical adsorption and **c**, chemical adsorption of equal amount of sulfur-adatoms on tellurium along (010) surface. The blue isosurface represents injection of electrons, and red for stripping or withdrawal of electrons. For clarity, the iso-charge contours are only shown in the right halves of **b** and **c**. Plane-integrated charge transfer along the surface normal direction are shown for both cases and are plotted with the same scale, so that they are quantitatively comparable.

Figure S19. High-resolution normalized XPS spectra of undoped Te nanowires (Te-PVP) in black and lightly and heavily doped nanowires (Te-PVP-S²⁻) in red and blue respectively. **The spectra are normalized to a peak value of 10 in arbitrary units to demonstrate the shift in peaks.** The presence of a sulfur peak around 169 eV in undoped Te nanowires as well suggests oxidized sulfur impurities in the sample. The gradual red shifting of the peak at 169 eV with increased doping is due to reduction of the oxidized sulfur impurities. A new peak around 162-162.5 eV is observed due to negatively charged sulfur-species (S²⁻).

XPS results show that there are some sulfur impurities in the as-synthesized Te-PVP sample either coming from substrate or in the sample itself. These sulfur atoms are not charged - black curve has no peak at 162 eV. So these are S⁰ species. As we add sulfur to the surface, we generate charged Sulfur species on the surface due to charge transfer from Te to sulfur – also generating the S-dopant band as well as localizing the electrons on the Surface layer (thus leading to charged S-species). Please note that **the spectra are normalized to observe the effective shift of the spectra. The actual amount of sulfur in the undoped samples is negligible** (see Figure below) – spectra are taken overnight to get a reliable signal.

Figure S18. Energy Dispersive X-ray Analyses spectra acquired during SEM imaging, indicating presence of both Tellurium from the nanowires and Sulfur on the surface. A weak almost negligible sulfur peak in the undoped Te nanowires (Te-PVP) could be due to sulfur impurities from the substrate or the sample while a much stronger peak is observed in the doped nanowires (Te-PVP-S²⁻). The Si peak is due to the substrate. The lack of an identifiable Na peak indicates effective removal of Na₂S or unbound S²⁻ ions.

Figure S20. (a, b) High-resolution XPS spectra of the Te-3d peak of undoped Te nanowires (Te-PVP) in black and doped nanowires (Te-PVP-S²⁻) in red for two different samples. In both samples, the TeO₂ peak is suppressed with doping proving that S²⁻ dopants passivate the nanowire surface effectively thus reducing oxidation. A slight blue shift to higher energies or an increase in binding energies can be observed in the zoomed-in images (c, d) for the doped samples which is suggestive of *p*-type doping

Also we observe a gradual shift in the tellurium peaks to more p-type which corroborates with the DFT results of charge transfer from Te to Sulfur.

4) **Work function (experimental) measurements demonstrating the reduction in workfunction by 0.3 eV towards conduction band** – The band gap of Te is 0.35 eV which implies that the work-function lies either in the dopant band or pretty close to the conduction band of Te.

Material	WF (eV)	IE (eV)
Te-PVP	4.0	4.6
Te-S ²⁻	3.7	4.1

Table 1. Experimental Work function (WF) and Ionization Energy (IE) for Te-PVP and Te-S²⁻ NW films as determined by Ultraviolet Photoelectron Spectroscopy showing the transition of the Fermi level from the valence band towards the conduction band due to sulfur doping.

Figure S24. Cartoon demonstrating the shift in work function due to sulfur doping (work function values are taken from Table 1 in main text). With a band gap of 0.35 eV for bulk Te, the change in work function by 0.3 eV implies that the Fermi level of the S-doped Te NWs lies either in the dopant band or very close to the conduction band of Te.

5) Transistor measurements that show a shift in the Fermi level gradually from close to the valence band to the conduction band as a function of increased dopant addition – corroborating the work function shift.

Also the lack of hysteresis in sample proves that Sulfur on the surface is effectively passivating the surface and filling in all surface states (indirect way of proving that sulfur is on the surface). Additionally, the change in nature of the transistors from p-type to ambipolar to n-type as a function of increasing sulfur concentration is a direct indication of the Fermi level moving closer from the valence band of tellurium (p-type) to somewhere within the band gap (ambipolar) ultimately close to the conduction band (n-type).

Figure 5. Fermi level shifts with doping. **a**, Schematic cross-section (not to scale) of ion-gel-gated thin-film transistors used to characterize the electrical properties of the doped NWs. The length and width of the channel were $100 \mu\text{m}$ and 2 mm , respectively. Red and blue circles represent positive and negative ions, respectively. **b**, Energy level diagram depicting the relationship between the location of the Fermi energy (E_F), band-edges, doping concentration and corresponding nature of charge carriers. CB and VB refer to the conduction band and the valence band of Te respectively. Transfer characteristics for the **c**, undoped Te-PVP NW sample (with $V_D = 0.1 \text{ V}$) showing p-type transport, **d**, intermediate-doped Te NWs ($\sim 1.5\%$ atomic concentration of sulfur, with $V_D = -0.3 \text{ V}$) showing ambipolar transport. **e**, heavily-doped Te NWs ($\sim 2.4\%$ atomic concentration of sulfur, with $V_D = -1.5 \text{ V}$) showing n-type transport. Black and red curves plot the characteristics on logarithm and linear scales respectively.

6) Seebeck coefficient change – gradually from positive values to negative values through experimental results – again corresponding to the calculated Seebeck coefficient values from DFT calculations with S-addition.

Experimental results:

Figure 4. Tuning Seebeck coefficients with controlled doping. a, Seebeck coefficient from a series of doped Te NW samples versus the amount of S^{2-} added to the exchange solution, normalized to the total number of Te atoms present on the surface of the NW. Insets show a representative set of smooth drop-cast films used for the measurements.

Simulated results using DFT and Boltzmann Transport Equations:

Figure S16. (a) Band structure of sulfur-doped tellurium where the sulfur is chemically adsorbed on the surface of tellurium obtained with GGA as implemented in the PAW scheme using VASP **(b)** Calculated density of states (DOS) **(c)** Model depicting the variation in Seebeck coefficient as a function of the Fermi level (E_F) taking into account both conduction (with effective mass $m_e^* = 0.06m_0$, where m_0 is the mass of free electron) and valence bands ($m_e^* = 0.114m_0$). E_V denotes the valence band edge.

What the figure is indicating is that based on the transport measurements (work function + transistor measurements), if the Fermi level is close to the conduction band of Te (say around 0.3 eV from the top of the valence band), the calculated Seebeck coefficient (based on the band structure from DFT calculations) will be nearly $-400 \mu\text{V/K}$ which is pretty close to the values of $-370 \mu\text{V/K}$ we observe experimentally. Please note that these calculations are performed on a single surface slab of Te and assume full surface coverage with S-atoms. Still, the theoretical and experimental values qualitatively match with each other.

The atom-projected band structure reveals conductive surface states that originate from the S-S chains along the surface in the physical adsorbed scenario due to the sulfur band crossing the Fermi level. Hence, the charge mobility should be increased greatly around the Fermi level. While, the chemical adsorbed structure does not have the conductive band crossing the Fermi level, it does introduce a new dopant band close to the conduction band edge which can explain the surprising n -type behavior of the sulfur-doped tellurium nanowires. If we compare the calculated Seebeck coefficients from the DOS for the three different cases cited above – undoped Te and sulfur-doped Te with sulfur adsorbed either physically or chemically (Figure S17), we can observe very distinct behavior for the variation in Seebeck coefficient as a function of the Fermi level in each of the three scenarios. In the case where sulfur is physically adsorbed on the surface of tellurium, the S-S chains introduce a lot of surface states which result in switching to negative Seebeck coefficients by shifting the Fermi level to about 25 meV above the valence band (VB) edge of tellurium. Similarly, for the chemically adsorbed case, a Fermi level that is about 190 meV above the VB edge results in negative Seebeck coefficients as compared to 240 meV for undoped tellurium. What these results point out is that, if we assume that the Fermi level in the three different scenarios does not change (*i.e.* no extra charge carriers are added to the system by doping), then with a fixed Fermi level, it is possible to obtain n -type doping or n -type transport (negative Seebeck coefficients) simply by modifying the local band structure around the Fermi level. For example, if we pin the Fermi level for all three systems at say 230 meV above the VB edge, while we would obtain a Seebeck coefficient of $440 \mu\text{V/K}$ for the undoped Te (p -type), we would obtain Seebeck coefficients of $-76 \mu\text{V/K}$ and $-680 \mu\text{V/K}$ (n -type) for the physi-sorbed and chemi-sorbed cases respectively. As discussed before, the two extreme cases of the sulfur-surface adsorptions, categorized as chemical-adsorption (or chemisorption) and physical-adsorption (or physisorption) exist with almost identical formation energy. At around room temperature, due to perturbations from thermal energy, in all likelihood, the real physical system might manifest as an intermediate case incorporating both chemically and physically adsorbed sulfur and thus, the Seebeck coefficients that we observe would be some intermediate value.

Figure S17. (a) Model depicting the variation in Seebeck coefficient as a function of the Fermi level (E_F) taking into account both conduction (with effective mass $m_e^* = 0.06m_0$, where m_0 is the mass of free electron) and valence bands ($m_e^* = 0.114m_0$). E_V denotes the valence band edge. Comparison between undoped tellurium and doped tellurium with chemically adsorbed and physically adsorbed sulfur. **(b)** Zoomed-in figure of (a) focusing on the region in around the valence band edge.

7) Temperature dependent electrical conductivity data – that demonstrates a switch in transport mechanism from localized transport within the narrow dopant band (dominated by scattering) to transport within the conduction band of tellurium (activated transport). The switch happened nominally at 225 K ($\sim 19\text{-}20$ meV) that matches up qualitatively with the location of the shallow dopant band right beneath the Te-conduction band (work function difference of nearly 50 meV).

Figure 6. Thermoelectric transport properties in surface re-engineered tellurium nanowires. **a**, Temperature dependent conductivity (σ) measurements (from 90 K – 225 K) on thin films of undoped *p*-type (Te-PVP, blue spheres) and sulfur-doped *n*-type (Te-S²⁻, red spheres) tellurium nanowires normalized to the respective conductivity values at 90 K (σ_{90K}). While the undoped nanowire films show activated (semiconductor-like) transport with conductivities increasing with increasing temperature, the doped nanowire films demonstrate band-like transport with decreasing values of conductivities with increasing temperature. **b**, Proposed doping scheme wherein sulfur dopes tellurium *n*-type and most of the electrons are located in the dopant band with the Fermi level inside the band (similar to a metal) at low temperatures (below 225 K) and hence the band-like transport behavior. Above 225 K, the electrons gain sufficient thermal energy to get promoted to the conduction band. **c**, Temperature dependent conductivity measurements for the *n*-type (Te-S²⁻) samples wherein above 225 K the conductivity increases sharply with increasing temperature

1. The authors are not adequately accounting for all that is occurring upon S²⁻ doping that could be influencing the material properties. A large amount of TeO₂ is apparent in the samples prior to S²⁻ doping, as shown in both XPS and XRD data, but after doping the TeO₂ signature is completely eliminated from the XRD data and the TeO₂ peaks in the XPS data are reduced. TeO₂ is a wide bandgap semiconductor and is *p*-type. If this TeO₂ is more localized at the NW surfaces, which seems reasonable, it should be influencing transport between nanowires. Elimination of part or all of this high bandgap TeO₂ alone may have a huge influence on electronic properties.

Response - We appreciate the Referee's concerns and apologize for the confusion. Freshly synthesized Te nanowires do not have an oxide layer – only after a few days/weeks of exposure to air do the surface of these nanowires oxidize to generate TeO₂ on the surface. Please find below plots of freshly synthesized Te nanowires and those after oxidation –

Figure S3: X-ray diffraction patterns for undoped Te NWs that are freshly prepared (red) and stored in ambient conditions for a few weeks (red). The pink bars denote the reference peaks for pure Te while the asterisks show peaks from TeO₂.

In contrast, once these freshly synthesized unoxidized Te NWs are coated with sulfur – even after weeks/months – they do not oxidize – thus we believe strongly that TeO_2 does not play any role in any of the observed n-type properties.

Figure S2. (b) X-ray diffraction patterns for undoped NWs (black) and doped NWs (red – 1.2%, blue -1.5% and green -2.4% in increasing order of dopant concentration). The purple bars denote the reference peaks for pure Te while the asterisks show peaks from TeO_2 . Sulfur doping prevents surface oxidation

XPS is extremely sensitive to the surface and hence will pick up even slight amounts of oxidation on the surface but is not representative of the entire sample – for example – these nanowires are pretty thick – to the tune of 70-80 nm. XPS is probing only the top 2-nm of these wires and is highly exaggerating the amount of TeO_2 . In XRD for the same samples, we cannot even pick up any TeO_2 signal. The XPS is performed only for freshly prepared Te samples that show minimal oxidation and Te-S^{2-} samples. Note that the intensity of Te-peaks is normalized to show the relative change in the oxidized peak.

Figure S20. (a, b) High-resolution XPS spectra of the Te-3d peak of undoped Te nanowires (Te-PVP) in black and doped nanowires (Te-PVP-S²⁻) in red for two different samples. In both samples, the TeO₂ peak is suppressed with doping proving that S²⁻ dopants passivate the nanowire surface effectively thus reducing oxidation. A slight blue shift to higher energies or an increase in binding energies can be observed in the zoomed-in images (c, d) for the doped samples which is suggestive of *p*-type doping.

Even freshly prepared Te nanowires without any TeO₂ on the surface are *p*-type and surface oxidation does not adversely affect the thermoelectric properties of the *p*-type Te nanowires (we have probed the Seebeck coefficients/electrical conductivity) of these nanowires over time and they do not change a lot with time (~10% at the most). The XRD was shown to demonstrate the fact upon months of exposure to air, while bare Te nanowires will oxidize, the Te-S²⁻ samples do not oxidize to a large extent and are hence much more robust.

We corroborate this using transistor measurements where slight surface oxidation of uncoated Te NWs might be leading to surface traps and hence hysteresis in the transport characteristics (Figure 5c). Once coated with Sulfur (Figure 5e), we have hysteresis free transfer curves which indicate the lack of any traps (most likely surface traps – since that is the only aspect of the wires we are changing).

Figure 5. Fermi level shifts with doping. **a**, Schematic cross-section (not to scale) of ion-gel-gated thin-film transistors used to characterize the electrical properties of the doped NWs. The length and width of the channel were $100 \mu\text{m}$ and 2mm , respectively. Red and blue circles represent positive and negative ions, respectively. **b**, Energy level diagram depicting the relationship between the location of the Fermi energy (E_F), band-edges, doping concentration and corresponding nature of charge carriers. CB and VB refer to the conduction band and the valence band of Te respectively. Transfer characteristics for the **c**, undoped Te-PVP NW sample (with $V_D = 0.1 \text{V}$) showing p-type transport, **d**, intermediate-doped Te NWs ($\sim 1.5\%$ atomic concentration of sulfur, with $V_D = -0.3 \text{V}$) showing ambipolar transport. **e**, heavily-doped Te NWs ($\sim 2.4\%$ atomic concentration of sulfur, with $V_D = -1.5 \text{V}$) showing n-type transport. Black and red curves plot the characteristics on logarithm and linear scales respectively.

2. Building on point 1, the role of interfaces and transport between nanowires is largely neglected. The elimination/reduction of TeO_2 and PVP at the NW surfaces, as well as the highly crystalline surfaces, obtained with S^{2-} doping will improve electronic contacts between NWs. The electronic effects at these NW-NW contacts are not adequately addressed, and are expected to be significant.

Response – The Referee raises an interesting point. Unfortunately, as the Referee points out correctly, it is extremely hard to make an apples-to-apples comparison especially in a disordered system (semi-amorphous system) where on one hand you have hole transport in undoped Te nanowires through the valence band, while in the other case you have electron transport in a combination of dopant/conduction band. We expect the scattering mechanisms to be completely different in both cases. Can we isolate and quantify these effects? – Possibly – yes. The way we

are going about this is through single nanowire measurements – preliminary measurements demonstrate that while contact effects between nanowires significantly affect the electrical conductivity values, the Seebeck coefficients remain largely unchanged (both for the n-type and p-type nanowires). We have unpublished preliminary data from single nanowire measurements that show that we preserve the trends in single nanowires. However, we believe these results require more careful evaluation and investigation and are beyond the current scope of this manuscript. Also, our Seebeck measurements are expected to be independent of the identify of metal contacts, since we apply a temperature gradient and measure the open-circuit voltage as a function of increasing gradient and then we reverse the direction of the gradient and measure the voltage again in the reverse direction. We ensure that at 0 temperature gradient, our open circuit voltages are 0; thus eliminating any possibility of metal-semiconductor contact effects at the junction.

Figure S11. Open circuit voltage versus applied temperature gradient for (a) polyvinylpyrrolidone-capped and (b) fully S²⁻-exchanged Te NW films. The Seebeck coefficient is derived from the slope of the linear fit (R^2 values of 0.9999 and 0.9996 respectively). Error bars representing the standard deviation from averaging 10 readings for each temperature gradient are captured within the data marker.

3. The authors propose that the S atoms remain near the surface, which is a major selling point of their doping strategy, but there is no evidence supporting where the S atoms are located. EDS line scans, EELS imaging, etc. is necessary to show whether the S atoms actually remain surface localized.

We thank the reviewer for the constructive comment and hence we performed the line scans as suggested and have now incorporated in the manuscript to substantiate our claims.

The high-angle annular dark-field scanning transmission electron microscopy (HAADF-STEM) and energy-dispersive X-ray (EDX) maps were performed on a FEI TitanX 60-300 microscope equipped with Bruker windowless EDX detector at an acceleration voltage of 60 kV. The sample was prepared by depositing a drop of dilute nanowire dispersion in water onto a lacy carbon 400-mesh copper grid placed upon a hot plate covered by clean filter paper. The film then was gently heated to 60 °C to allow the solvent to evaporate. The STEM-EDX and EDX line scans were used to analyze the elemental distribution in S doped Te nanowires. The images clearly show that sulfur is concentrated in the surface region, while Te is concentrated in the core region.

Figure 2. Structural characterization of dopant location. a, high-angle annular dark-field scanning transmission electron microscopy (HAADF-STEM) and b, c, energy-dispersive X-ray (EDX) maps of sulfur and tellurium respectively in a sulfur-doped tellurium nanowire showing the presence of a minute amounts of sulfur in the sample d, line scans demonstrating that sulfur atoms are primarily concentrated on the surface of the nanowire.

Figure S4. STEM EDX spectrum before and after deconvolution: The spectrum shows the presence of a tiny amount of Sulfur in the S-doped Te samples. The approximate value from the spectrum is S: 0.38 at.%, Te: 99.62 at.%. However, since the S dopant concentration is too low, and the EDX data generates a systematic error around 1-2% for quantification considering the errors in background subtraction, data fitting etc., thus the absolute quantification value for S and Te are not accurate.

4. The presence of a large amount of S in the undoped samples is concerning. The S peak in the XPS sample for the undoped Te NWs is much more intense than the peak from the S²⁻ in the doped samples. It would be helpful if the XPS data was used to calculate atomic concentrations to provide a gauge on the amount of S present in all samples.

We are slightly confused by the comment. There is negligible amount of sulfur as is evident from the EDS scans below and the STEM/EDX data shown above. We gather the Reviewer is concerned about the Sulfur present in undoped Te NWs. The XPS scans were normalized to demonstrate the shift in S-peak and are not representative of the actual S-concentration. Also, since XPS probes only the top 2-3 nm of the sample, it will be incorrect to calculate atomic percentages based on XPS data since we will not be probing the bulk of the Tellurium thus giving us spuriously high values of Sulfur in our samples.

Figure S19. High-resolution normalized XPS spectra of undoped Te nanowires (Te-PVP) in black and lightly and heavily doped nanowires (Te-PVP-S²⁻) in red and blue respectively. The spectra are normalized to a peak value of 10 in arbitrary units to demonstrate the shift in peaks. The presence of a sulfur peak around 169 eV in undoped Te nanowires as well suggests oxidized sulfur impurities in the sample. The gradual red shifting of the peak at 169 eV with increased doping is due to reduction of the oxidized sulfur impurities. A new peak around 162-162.5 eV is observed due to negatively charged sulfur-species (S²⁻).

XPS results show that there are some sulfur impurities in the as-synthesized Te-PVP sample either coming from substrate or in the sample itself. These sulfur atoms are not charged - black curve has no peak at 162 eV. So these are S⁰ species. As we add sulfur to the surface, we generate charged Sulfur species on the surface due to charge transfer from Te to sulfur – also generating the S-dopant band as well as localizing the electrons on the Surface layer (thus leading to charged S-species). Please note that **the spectra are normalized to observe the effective shift of the spectra. The actual amount of sulfur in the undoped samples is negligible** (see Figure below) – spectra are taken overnight to get a reliable signal.

Figure S16. Energy Dispersive X-ray Analyses spectra acquired during SEM imaging, indicating presence of both Tellurium from the nanowires and Sulfur on the surface. A weak almost negligible sulfur peak in the undoped Te nanowires (Te-PVP) could be due to sulfur impurities from the substrate or the sample while a much stronger peak is observed in the doped nanowires (Te-PVP-S²⁻). The Si peak is due to the substrate. The lack of an identifiable Na peak indicates effective removal of Na₂S or unbound S²⁻ ions.

5. Measurements of charge carrier densities in these materials, such as through Hall effect measurements, are necessary for understanding the mechanism of S²⁻ treatment. Carrier densities are discussed, but no direct measurement of these carrier densities are presented. Measurements of carrier densities are particularly important given that S²⁻ doping will simultaneously influence both the electronic contact between NWs and the carrier densities.

We whole-heartedly agree with the Reviewer that charge carrier density measurements will be extremely useful. In disordered nanocrystalline systems, it is hard to calculate/estimate carrier concentrations from Hall measurements owing to anomalous Hall effects (drift diffusion is not valid anymore – hence giving anomalous Hall voltages). Thus we cannot extract Hall mobility and hence carrier concentrations. Other way to calculate carrier concentrations would be through field effect transistor measurements – unfortunately in low band gap materials with intrinsically high carrier concentrations, the current modulation achieved through field effect gating is too low to get a reliable number for mobility and carrier concentrations that is trustworthy. Hence we used Boltzmann transport calculations based on DFT results to validate observed experimental results (Supplementary Sections S3 and S5).

Smaller points:

In Figure S1 there is not much contrast between the crystalline Te and the amorphous polymer. C and Te should show a large amount of contrast and it is possible that this amorphous polymer layer is actually TeO₂.

We appreciate the Reviewer's comment and we certainly thought about this. There are two reasons we believe that the amorphous layer is not TeO_2

1) the thickness of the amorphous layer is not uniform and does not mimic the smooth surface of the crystalline wire as is observed from the TEM images; it has weird "soft" and "random" shapes and thicknesses. While possible, it is highly unlikely to fathom that the TeO_2 layer is growing as random structures on the surface of crystalline Te.

2) It is highly unlikely that addition of sulfur will etch away this thick TeO_2 layer and leave an extremely pristine interface of Te without any surface defects/amorphous layer. We can observe in the 2nd image that the S-doped Te nanowires are perfectly crystalline right up to the surface.

Figure S1. Preserving crystallinity of nanowires with doping – High resolution transmission electron micrographs of undoped Te nanowires capped with polymer (polyvinylpyrrolidone, PVP) and sulfur doped Te nanowires. An amorphous polymer (PVP) layer on the surface of the tellurium nanowire with thicknesses typically on the order of 2-5 nm can be observed. In the sulfur-doped Te nanowires, the crystalline Te domain extends upto the edge of the nanowire and no amorphous layer is observed suggesting that all the polymer has been removed.

Ultraviolet photoelectron spectroscopy measurements to compare to calculated band structures and to directly probe the difference between the Fermi energy and the valence band onset in the different samples would be helpful.

We have performed UPS measurements and we included it in Table 1 in the main text of the manuscript:

The band gap of Te is 0.35 eV which implies that the work-function lies either in the dopant band or pretty close to the conduction band of Te.

Material	WF (eV)	IE (eV)
Te-PVP	4.0	4.6
Te-S ²⁻	3.7	4.1

Table 1. Experimental Work function (WF) and Ionization Energy (IE) for Te-PVP and Te-S²⁻ NW films as determined by Ultraviolet Photoelectron Spectroscopy showing the transition of the Fermi level from the valence band towards the conduction band due to sulfur doping.

Since these values strongly depend on the surface of the nanowires and there will be sample inhomogeneities in the amount of sulfur of the surface and leftover PVP (and consequently will drastically affect the work function and ionization energy values), we only use the numbers to demonstrate a qualitative effect of the surface modification on the WF and IE. We find that both values decrease which demonstrate that it is now easy to extract electrons from the surface which is akin to doping a material n-type and the work function moving to lower values is a clear indication that the Fermi level in the film moves towards the conduction band. We corroborate these results with the transistor measurements that also demonstrate a clear trend of the Fermi level shifting towards the conduction band with increased sulfur concentration the surface of the nanowires. Please find below a schematic that shows these trends as a function of doping. We have included the figure in the SI as well.

Figure S25. Cartoon demonstrating the shift in work function due to sulfur doping (work function values are taken from Table 1 in main text). With a band gap of 0.35 eV for bulk Te, the change in work function by 0.3 eV implies that the Fermi level of the S-doped Te NWs lies either in the dopant band or very close to the conduction band of Te.

The temperature dependent conductivity measurements of the doped samples are interesting, particularly the change in activation energy occurring after 225 K. Seebeck measurements over this entire range would be helpful in interpreting what is occurring, but Seebeck measurements are only presented with temperatures above 225K.

We thank the Referee for the useful suggestion: temperature dependent Seebeck coefficient measurements would certainly be helpful and would shed more light on the observed effects and interpret exactly what is going on in the samples. Unfortunately, Seebeck coefficient measurements are only performed for room temperature and higher temperatures (not 225 K and above). Qualitatively the rise in Seebeck coefficients makes sense – if the Fermi level is in the dopant band, and we increase the temperature, the Fermi window is going to smear out of the dopant band. Now below the dopant band we do not have any states (in the band gap). However, above the dopant band, we have the conduction band. Thus, with increase in temperature – we will have more and more states contributing to electron transport rather than hole transport. This asymmetry in the density of states with increase in temperature will lead to an increase in the negative value of Seebeck coefficient as we qualitatively observe from room temperature and higher temperatures.

Figure 6. Thermoelectric transport properties in surface re-engineered tellurium nanowires.
e, Temperature dependent Seebeck coefficient of the *n*-type Te nanowire film.

We do not have the apparatus to perform low Temperature Seebeck coefficient measurements. However, the calculations of Seebeck coefficients using Boltzmann Transport equations which were performed on the DFT-calculated band structure and the anticipated location of the Fermi level – matches well with the observed values of the Seebeck coefficient. We have extensively discussed the various cases in the SI as demonstrated below.

S3.3) Band Structure Calculations using DFT – Origin of *n*-type transport

Figure S14. (a) Band structure of tellurium obtained with GGA as implemented in the PAW scheme using VASP (b) Calculated density of states (DOS) (c) Model depicting the variation in Seebeck coefficient as a function of the Fermi level (E_F) in Te, taking into account both conduction (with effective mass $m_e^* = 0.06m_0$, where m_0 is the mass of free electron) and valence bands ($m_e^* = 0.114m_0$). E_V denotes the valence band edge.

Figure S15. (a) Band structure of sulfur-doped tellurium where the sulfur is physically adsorbed on the surface of tellurium obtained with GGA as implemented in the PAW scheme using VASP **(b)** Calculated density of states (DOS) **(c)** Model depicting the variation in Seebeck coefficient as a function of the Fermi level (E_F) taking into account both conduction (with effective mass $m_e^* = 0.06m_0$, where m_0 is the mass of free electron) and valence bands ($m_e^* = 0.114m_0$). E_V denotes the valence band edge.

Figure S16. (a) Band structure of sulfur-doped tellurium where the sulfur is chemically adsorbed on the surface of tellurium obtained with GGA as implemented in the PAW scheme using VASP (b) Calculated density of states (DOS) (c) Model depicting the variation in Seebeck coefficient as a function of the Fermi level (E_F) taking into account both conduction (with effective mass $m_e^* = 0.06m_0$, where m_0 is the mass of free electron) and valence bands ($m_e^* = 0.114m_0$). E_V denotes the valence band edge.

Figure S17. (a) Model depicting the variation in Seebeck coefficient as a function of the Fermi level (E_F) taking into account both conduction (with effective mass $m_e^* = 0.06m_0$, where m_0 is the mass of free electron) and valence bands ($m_e^* = 0.114m_0$). E_V denotes the valence band edge. Comparison between undoped tellurium and doped tellurium with chemically adsorbed and physically adsorbed sulfur. **(b)** Zoomed-in figure of (a) focusing on the region in around the valence band edge.

How were the WFs and IEs measured?

We thank the reviewer for pointing out this oversight. We apologize for the confusion since we did not clearly mention in the previous version of the manuscript whether these were calculated or experimental values. Both WF and IE values are experimentally extracted from UPS spectra. The procedure (described below) has now been added to the SI to increase the clarity of the manuscript: Ultraviolet photoelectron spectroscopy (UPS) was performed on a Thermo K-Alpha Plus instrument using a He I source (21.2 eV), and values for the work function (WF) and ionization energy (IE) for the Te-PVP and Te-S²⁻ NW film were extracted via UPS spectra. All thin film surfaces were cleaned using an Ar cluster gun (6000 eV/cluster, 150 atoms/cluster, 15 sec) before all UPS experiments. Work functions were extracted as the difference between the UPS radiation source energy and the secondary electron cutoff energy, whereas ionization energies were calculated by measuring the valence band onset.

Changes in the SI:

Ultraviolet photoelectron spectroscopy (UPS) was performed on a Thermo K-Alpha Plus instrument using a He I source (21.2 eV), and values for the work function (WF) and ionization energy (IE) for the Te-PVP and Te-S²⁻ NW film were extracted via UPS spectra. All thin film surfaces were cleaned using an Ar cluster gun (6000 eV/cluster, 150 atoms/cluster, 15 sec) before all UPS experiments. Work functions were extracted as the difference between the UPS radiation source energy and the secondary electron cutoff energy, whereas ionization energies were calculated by measuring the valence band onset.

Assuming mobilities of 2380 cm²/Vs based on bulk Te, or perhaps single crystal Te (I could not access the reference to find out), is not a fair assumption. NWs, and particularly NW films, display much lower mobilities.

We agree with the Referee that it is not a fair assumption but without a reliable technique to calculate carrier mobility and charge carrier concentrations the best we could do was to come up with a back of an envelope calculation and surprisingly, our numbers match up very well with observed experimental results –

“Referring back to our earlier predictions regarding the origin of the *n*-type conductivity, where we estimated that even at 2.5% concentration of dopant, the surface-bound sulfur dopants can provide 1000 times more electrons (assuming 100% carrier-doping efficiency) as compared to intrinsic hole concentrations (4.8×10^{17} carriers/cm³). Thus, our estimations indicate carrier concentrations in the doped nanowires ought to be nearly 4.8×10^{20} electrons/cm³. With this carrier concentration and an activation barrier energy of 336 meV (obtained from the temperature-

dependent conductivity studies), we estimate the concentration of free carriers in the conduction band at 300 K to be nearly 7×10^{14} electrons/cm³. Assuming bulk electron mobilities (obtained from hall measurements at high temperatures due to promotion of electrons from valence to conduction band) of 2380 cm²/V.s,⁵⁸ we estimate the conductivity at 300 K to be nearly 0.3 S/cm respectively. Experimental value of conductivity for the same sample at 299 K is 0.6 S/cm which is consistent with the physical picture that the electrical conductivity in our *n*-type systems originate from the charge carriers that are promoted into the conduction band from the dopant band.”

As the referee correctly pointed out, lower mobility in nanostructures typically can occur due to defects in individual wires (surface or intrinsic). In assemblies of these wires, inter wire transport can dominate overall transport but within a single nanowire – that does not demonstrate any quantum effect – it can be of similar order as that of bulk. And that is exactly what we observe for measurements on single nanowires. Our single nanowires in fact demonstrate electrical conductivity values that are higher than bulk Te (Unpublished Data - we are still probing the origins). However, the results clearly convey that we start off with a much higher electrical conductivity in individual nanowires – then owing to scattering of charges at interfaces during inter-wire transport, we reduce the electrical conductivity by a bit but still maintain reasonably high values of electrical conductivity. Most likely, in these long nanowires (10-30 μm), since we are measuring in-plane electrical conductivity while these are lying flat on the surface – the number of interfaces that charges encounter while traveling between electrodes is low and does not lead to drastic reduction in the overall film mobility.

Figure: Unpublished data for measurements on single nanowires. In our single suspended Te nanowires, electrical conductivity (~10 S/cm) is higher than bulk polycrystalline Te films (~3 S/cm).

More credit should be given to other solution processed thermoelectric materials. Currently, the manuscript focuses on comparison to organics, but many inorganic materials are also able to be solution processed. For example, Nature Energy volume 3, pages301–309 (2018) and Scientific Reports volume6, Article number: 33135 (2016).

We thank the reviewer for their suggestion and have added the references to the manuscript.

Reports utilizing solution-processed nanomaterials without additional energy intensive (and costly) post-processing, such as spark-plasma-sintering (SPS), are limited.⁴³⁻⁴⁹

(48) Kim, F.; Kwon, B.; Eom, Y.; Lee, J. E.; Park, S.; Jo, S.; Park, S. H.; Kim, B.-S.; Im, H. J.; Lee, M. H.; Min, T. S.; Kim, K. T.; Chae, H. G.; King, W. P.; Son, J. S. *Nature Energy* **2018**, *3*, 301.

(49) Varghese, T.; Hollar, C.; Richardson, J.; Kempf, N.; Han, C.; Gamarachchi, P.; Estrada, D.; Mehta, R. J.; Zhang, Y. *Scientific Reports* **2016**, *6*, 33135

However we would like to point the key difference between these articles and our manuscript – For example – in the Nature Energy paper - these inorganic materials are not solution-processed in the true sense – these are simply bulk materials that have been micro-grained using high energy ball milling processes (energy intensive process), then sieved to only retain the smallest grains (high volume of material wastage) and are in a metastable suspension (partially stable – hence need to be used as soon as possible or else need to be mixed/vortexed for hours). In contrast, our samples do not employ any such processes and are stable in the ambient in solution for months to years – thus our comparison to organics which are truly solution processable.

Nature Energy paper – “*Synthesis of all-inorganic Bi₂Te₃-based ink. The p-type and n-type TE powders were prepared with the stoichiometric composition of Bi_{0.4}Sb_{1.6}Te₃ and Bi₂Sb_{2.7}Se_{0.3}, respectively, by high-energy ball milling of Bi, Sb, Te and Se for 5h to produce grain sizes smaller than 45µm. Agglomerated particles were removed by sieving the powders to a particle size. A mixture of 2 g of the TE powder and the desired amount of ChaM were dispersed in 2 g of glycerol, and the solution was mixed with a planetary centrifugal mixer (ARM-100, Thinky) for 2h to fully homogenize the ink. Five zirconium oxide grinding balls of 5mm diameter were added to expedite the homogenization process. The whole synthesis process was carried out under a N₂ atmosphere.*”

The figures in the SI should be numbered in order of when they are mentioned in the text.

We have tried to organize the SI in terms of sections (electrical/structural characterization, DFT calculations) that are coherent – even though they come in a different order as mentioned in the text. In light of the Reviewer’s comments, we have re-organized some of the figures to be in line with the order they are introduced in the main text.

Initial XRD should be shown for the undoped Te NWs, as the XRD data in Figure S2 is after weeks of exposure to ambient atmosphere.

We agree with the reviewer and have now added the initial XRD of the undoped Te NWs that are freshly prepared.

Figure S3: X-ray diffraction patterns for undoped Te NWs that are freshly prepared (red) and stored in ambient conditions for a few weeks (red). The pink bars denote the reference peaks for pure Te while the asterisks show peaks from TeO_2 .

Reviewer #2 (Remarks to the Author):

In this paper, the authors report a surface-engineered doping strategy for tellurium nanowire by sulfur anion ligands in which p-type properties were dramatically changed to n-type. To verify the effectiveness of this approach, the authors systematically analysed structural and electrical changes of tellurium nanowire as increasing doping concentration of sulfur. Their fundamental study is well organized, and the following results are also very impressive. Furthermore, the authors successfully produced thin film transistors to demonstrate its generality over other electronic devices, and flexible thermoelectric generator to show its excellent applicability. Accordingly, I recommend the publication of this manuscript in Nature Communications since the subject matter can potentially appeal to a broad audience in the thermoelectric society. Some minor comments are as follows.

We thank the Reviewer for their positive endorsement of our work.

1. What is the possible range of doping? The authors need to report on the doping limits of sulfur for tellurium.

We discuss the limits of doping in Supplementary Section S4 wherein we present the elemental analyses data. The doping limits are governed by the surface to volume ratio of the nanowires.

“Since the average diameter of our Te nanowires is nearly 80-nm, only about 2.5% of the total Te atoms constitute the surface. Assuming 100% coverage of the surface Te atoms with S²⁻ atoms would give us only about 2.5% S-species in the samples. While it remains a challenge to accurately quantify the S²⁻ incorporation at low concentrations, qualitatively we are able to use electron dispersive spectroscopy (EDS) and X-ray photoelectron spectroscopy (XPS) to observe a general increase in sulfur concentration with increasing dopant addition. For fully surface-exchanged samples, while quantifying by EDS gives us nearly 2.4% sulfur, quantification by XPS gives us nearly 2.2% sulfur. These values are pretty close to what one would expect with complete surface exchange (2.5%).”

2. I wonder if the authors have tried to use TE nanowire with different diameter? Since this strategy is strongly related to surface chemistry, the surface to volume ratio of Te nanowires could influence to doping effect.

We have tried Te nanowires of different diameters (~10-nm) and have included the results in the Supplementary Section.

Figure S23. (a) Seebeck coefficient from a series of doped Te nanowire samples versus the amount of S^{2-} added to the exchange solution, normalized to the total number of Te atoms present on the surface of the nanowire. This data is obtained from a different sample batch than Fig. 4a of the main text with ~ 10 -nm diameter Te nanowires. Error bars represent the standard deviation from Seebeck coefficient measurements for each sample and at least 3 samples for each doping concentration. (b) Model depicting the variation in Seebeck coefficient (black curve) as a function of the position of the Fermi level (E_F) in Te. The red curve represents the parabolic band structure used in modeling, with CB and VB referring to the conduction band and the valence band respectively, with E_{BG} ($= 0.335$ eV) as the band gap and E_C refers to the conduction band edge of bulk Te. (c) Zoomed-in area of (b) around the band gap, rotated by 90° so the Fermi level is the x-axis. The shaded area in blue is the region depicted in (a) with the dotted black line denoting the Fermi level for undoped Te nanowires. The Fermi level can be shifted by gradual doping with S^{2-} . The surface area of the nanowires limits the maximum number of dopants that can be added and hence the region in purple depicts the inaccessible region of the band structure of Te.

Figure S24. (a) Electrical conductivity from the same series of doped Te nanowire samples as shown in Figure S23 versus the amount of S²⁻ added to the exchange solution, normalized to the total number of Te atoms present on the surface of the nanowire. Error bars represent the standard deviation from electrical conductivity measurements for each sample and at least 3 samples for each doping concentration. We observe a decrease in the conductivity at low doping levels and then a steady increase with higher doping amounts which corresponds with our hypothesis that the Fermi level shifts from a level close to the valence band towards the conduction band with doping. (b) Model depicting the variation in conductivity (black curve) as a function of the position of the Fermi level in tellurium. The overlaid red curve represents the parabolic band structure used in modeling, with CB and VB referring to the conduction band and the valence band respectively, with E_{BG} (= 0.335 eV) as the band gap and E_C refers to the conduction band edge of bulk Te. The shaded area in blue is the region depicted in (a).

3. Identifying the doping possibility for other isovalent elements may help clarify the doping mechanisms suggested in this manuscript. (i.e. Sulfur doping for selenium or selenium doping for tellurium.)

We thank the Referee for the useful suggestion. We have tried using Selenium instead of Sulfur but unfortunately we did not see any radical changes in the properties of Tellurium. We believe this might be due to the fact that Se has a lower electronegativity than S and hence it is probably forming mid-gap states in the gap (instead of a resonant band close to the conduction band of Te) or is not strong enough to pull electrons from Te into the Se-layer to invert the carrier type. DFT calculations to probe the electronic structure of the Te-Se combination can most likely provide more insight as to whether the Se can induce a similar resonant band in Te as Sulfur does. We strongly believe that the Te-S combination is not unique and many such host-dopant combinations ought to exist. High-throughput calculations can help identify such systems and provide a guideline for experimentalists to pursue ideal combinations.

Reviewer #3 (Remarks to the Author):

This work reports on the modulation of electrical transport and thermoelectric properties in Te nanowires by interfacial resurfacing/doping with S. According to the authors, there are mainly two effects of S-doping: one is the shift of Fermi level, and the other is the formation of a new, dopant band. In this manuscript, to my feelings, the two effects are somewhat mixed up. It is not clear whether the doping effects are body effects (like conventional effects in bulks) or interface effects. The key phenomenon of p-n conversion is not clearly understood. It is reasonable that after doping, S is partially negative and Te is positive, but this cannot explain why the system as a whole is n-type. Also, the resonant dopant band lacks substantiation.

Response: We agree with the Reviewer that it is infact a convoluted effect – in most cases, scientists observe either one or the other behavior. For us, we observe both in the same system which makes it interesting as well as challenging to probe experimentally and understand phenomenologically what is going on in the system. Simply put, it is an interface effect that leads to a formation of a new band. At low temperatures (<225 K), transport occurs through this narrow band (temperature-dependent conductivity measurements) and one observes scattering dominated transport. At higher temperature (>225 K) and at room temperature, the transport takes place through the conduction band of Tellurium due to thermally activated charges from the dopant band (it is strictly not a bulk effect – however similar effect can be realized in bulk – resonant doping Tl:PbTe etc). The system as a whole is n-type because of where the Fermi level is located and the local density of states around this fermi level. It might be simpler to think of this system not as individual S and Te (akin to lets say P-doped Silicon – we do not imagine this to P-states + Si bands) but a hybrid S:Te system where the Fermi level is within a dopant-induced band (contributed primarily by S-states) next to the conduction band (contributed primarily by Te-states).

Please refer to Comment#1 from Reviewer 1 wherein we justify the effect of the resonant band on the n-type thermoelectric properties of tellurium.

In addition, the device efficiency is not discussed, so the quality of the material and the device cannot be evaluated.

In this report, we do not emphasize the device performance. Rather, in this article, we explore a fundamental effect played by surface doping to generate resonant density of states (resonant dopant band) that completely inverts the charge carrier nature of a material. While effects of resonant doping exist that take p-type materials and improve the p-type properties, never before has this effect been exploited to change a p-type material to an n-type material. The approach lays the foundation to explore a wide range of material combinations to similarly discover, enhance and generate both electronic and TE materials with tunable material properties.

With regards to device efficiency, we believe the thin-film power factor is a good indicator of material properties. To use these nanowires in devices, we need to optimize casting conditions, thicknesses and loads, which is the subject of our future work. The unique aspect of these solution synthesized and solution processed materials is the absence of use of any hot pressing or sintering – thus rendering it amenable to be used on any form of substrates with flexible form factors. As far as device efficiencies go, preliminary measurements for the nano-systems give us

around 0.4 W/m.K for in-plane thermal conductivities and 0.2 W/m.K for through-plane. However, these measurements were conducted at room-temperature. In order to fully exploit the high thermoelectric power factor of these Te-nanowires, we need to conduct these measurements of thermal conductivity at higher temperatures where we obtain high power factors to get robust ZT values. Assuming we retain the same values of thermal conductivity at 430 K wherein we report a power factor value of nearly 500 $\mu\text{W}/\text{m.K}^2$, we would obtain a ZT value of ~ 0.54 which we believe is a decently high value for a fully solution-synthesized and solution-processed nanomaterial.

In addition, some concerns or comments on the technical details are also listed below.

1. As shown in Fig. 2 (d) and (e), there is little difference in band structure between undoped and S-doped Te except for the “dopant band” in the latter. Does the dashed line represent the Fermi level? If so, undoped Te (Fig. 2d) should also be n-type.

We apologize for the lack of clarity in the figure. The dotted black line denotes the edge of the conduction band so that it is easy to see where the CB is (the dashed line does not show the Fermi level). We had included this information in the figure caption. Please find below highlighted in yellow.

Figure 3. Density functional theory calculations showing charge transfer and tuning Seebeck coefficients with controlled doping. a, Illustration of the atomistic structure and (010) surface, of the hexagonal tellurium nanowire. The supercell configuration and the charge transfer effect for b, physical adsorption and c, chemical adsorption of equal amount of sulfur-adatoms on tellurium along (010) surface. The blue isosurface represents injection of electrons, and red for stripping or withdrawal of electrons. For clarity, the iso-charge contours are only shown in the right halves of b and c. Plane-integrated charge transfer along the surface normal direction are shown for both cases and are plotted with the same scale, so that they are quantitatively comparable. d, e, Calculated surface density of states using density functional theory (DFT) for tellurium (*p*-type) and sulfur-doped tellurium (*n*-type) with CB and VB referring to the conduction and valence bands of bulk tellurium respectively. **The dotted black line denotes the edge of the CB. A new dopant band (shaded ellipse) emerges close to the conduction band edge for the sulfur-doped tellurium.**

2. The evidence for the resonant band just from calculation is inadequate. Perhaps experimental Pisarenko relation (Seebeck coefficient vs carrier concentration) or optical measurements can be helpful.

We agree with the Reviewer that additional experimental results can lend weight to our conclusions but we believe we have provided enough evidence to justify our case (please refer to comments from reviewer 1). In disordered nanocrystalline systems, it is hard to calculate/estimate carrier concentrations from Hall measurements owing to anomalous Hall effects (drift diffusion is not valid anymore – hence giving anomalous Hall voltages). Thus we cannot extract Hall mobility and hence carrier concentrations. Other way to calculate carrier concentrations would be through field effect transistor measurements – unfortunately in low band gap materials with intrinsically high carrier concentrations, the current modulation achieved through field effect gating is too low to get a reliable number for mobility and carrier concentrations that is trustworthy. Hence we used Boltzmann transport calculations based on DFT results to validate observed experimental results (Supplementary Sections S3 and S5) qualitatively using Pisarenko relations. Instead of having carrier concentrations in the x-axes, we have the Fermi level which is a direct function of the Sulfur concentration.

Figure S23. (a) Seebeck coefficient from a series of doped Te nanowire samples versus the amount of S^{2-} added to the exchange solution, normalized to the total number of Te atoms present on the surface of the nanowire. This data is obtained from a different sample batch than Fig. 4a of the main text with ~ 10 -nm diameter Te nanowires. Error bars represent the standard deviation from Seebeck coefficient measurements for each sample and at least 3 samples for each doping concentration. (b) Model depicting the variation in Seebeck coefficient (black curve) as a function of the position of the Fermi level (E_F) in Te. The red curve represents the parabolic band structure used in modeling, with CB and VB referring to the conduction band and the valence band respectively, with E_{BG} ($= 0.335$ eV) as the band gap and E_C refers to the conduction band edge of bulk Te. (c) Zoomed-in area of (b) around the band gap, rotated by 90° so the Fermi level is the x-axis. The shaded area in blue is the region depicted in (a) with the dotted black line denoting the Fermi level for undoped Te nanowires. The Fermi level can be shifted by gradual doping with S^{2-} . The surface area of the nanowires limits the maximum number of dopants that can be added and hence the region in purple depicts the inaccessible region of the band structure of Te.

Figure S24. (a) Electrical conductivity from the same series of doped Te nanowire samples as shown in Figure S23 versus the amount of S^{2-} added to the exchange solution, normalized to the total number of Te atoms present on the surface of the nanowire. Error bars represent the standard deviation from electrical conductivity measurements for each sample and at least 3 samples for each doping concentration. We observe a decrease in the conductivity at low doping levels and then a steady increase with higher doping amounts which corresponds with our hypothesis that the Fermi level shifts from a level close to the valence band towards the conduction band with doping. (b) Model depicting the variation in conductivity (black curve) as a function of the position of the Fermi level in tellurium. The overlaid red curve represents the parabolic band structure used in modeling, with CB and VB referring to the conduction band and the valence band respectively, with E_{BG} ($= 0.335$ eV) as the band gap and E_C refers to the conduction band edge of bulk Te. The shaded area in blue is the region depicted in (a).

In low band gap materials, it is extremely hard to observe the band edge or any shift with optical measurements – we tried these measurements but did not find these to be conclusive because at these low energies (0.3 eV and lower), we observe significant scattering owing to the large sizes of these nanowires and also residual absorption from some of the leftover PVP (C-H stretches that absorb in these energy ranges). It was one of the primary reasons why we resorted to DFT calculations to corroborate our experimental observations with simulations. One possible route will be to conduct low temperature optical measurements that will reduce thermal noise and might allow to observe change in band gap and optical transitions more clearly. The other way would be directly probe the band structure using ARPES (Angle-Resolved Photoelectron Spectroscopy) – unfortunately ARPES is well-suited for single crystals. In a disordered polycrystalline system as ours, it will be hard to get robust data from ARPES. While interesting to try out, these additional experiments are not trivial and are beyond the scope of this manuscript. In our response to Reviewer 1, we thoroughly indicate how all our experimental observations agree with each other and point to the existence of the resonant band and its effect on the thermoelectric properties of Tellurium.

3. HR-TEM and/or other analyses are needed to show the actual position of S in Te matrix, i.e., substitutional, interstitial or precipitation, which is the basis to understand the change in electrical properties.

We thank the reviewer for the constructive comment and hence we performed the line scans as suggested and have now incorporated in the manuscript to substantiate our claims.

Figure 2. Structural characterization of dopant location. a, high-angle annular dark-field scanning transmission electron microscopy (HAADF-STEM) and b, c, energy-dispersive X-ray (EDX) maps of sulfur and tellurium respectively in a sulfur-doped tellurium nanowire showing the presence of a minute amounts of sulfur in the sample d, line scans demonstrating that sulfur atoms are primarily concentrated on the surface of the nanowire.

The high-angle annular dark-field scanning transmission electron microscopy (HAADF-STEM) and energy-dispersive X-ray (EDX) maps were performed on a FEI TitanX 60-300 microscope equipped with Bruker windowless EDX detector at an acceleration voltage of 60 kV. The sample was prepared by depositing a drop of dilute nanowire dispersion in water onto a lacy carbon 400-mesh copper grid placed upon a hot plate covered by clean filter paper. The film then was gently heated to 60 °C to allow the solvent to evaporate. The STEM-EDX and EDX line scans were used to analyze the elemental distribution in S doped Te nanowires. The images clearly show that sulfur is concentrated in the surface region, while Te is concentrated in the core region.

Figure S4. STEM EDX spectrum before and after deconvolution: The spectrum shows the presence of a tiny amount of Sulfur in the S-doped Te samples. The approximate value from the spectrum is S: 0.38 at.%, Te: 99.62 at.%. However, since the S dopant concentration is too low, and the EDX data generates a systematic error around 1-2% for quantification considering the errors in background subtraction, data fitting etc., thus the absolute quantification value for S and Te are not accurate.

4. Before discussing S-doped Te, the authors should make it clear why intrinsic Te is p-type. What are the intrinsic defects? Then how are the defects affected by S doping?

The reviewer raises an important question. Not much is known about the transport mechanism and intrinsic defects in Tellurium since it is not as heavily studied as other thermoelectric materials. Based on references back in the 1960's and 1970's, undoped Te is intrinsic at temperatures above 200 K. Extrinsic conductivity is of p-type, n-type crystals are not known.

For a systematic doping group V-elements are used (e.g. As-doped Te – *Nature Communications* 2016, 7, 10287), they act as shallow acceptors. Since not a lot of experiments have been performed with intrinsic samples except for one report (PhD Thesis, Fig. 99, *Baumgart, H. D.*:

Diplomarbeit, Universität Köln (1966)) that in fact reports a n-type Seebeck coefficient at room temperature for intrinsic Te (carrier concentration $\sim 6 \cdot 10^{14} \text{ cm}^{-3}$). However, most crystals have some degree of Group V impurities that lead to predominantly p-type character.

Since intrinsic Te does show n-type character owing to the high mobility of electrons which is greater than that of holes, thus it is plausible to imagine that it is in fact possible to derive n-type behavior from Te-crystals. Most likely common n-type dopants in Te (such as halide ions) might be forming deep defects or defect complexes which have led to no observation of n-type behavior – however without thorough calculations of defect levels and experimentation, we cannot say for sure why no n-type doping in Te has ever been reported.

Recent work by the group of Goddard et al. (J. Am. Chem. Soc. 2018, 140, 550–553) also suggests that Te might owe its p-type character because of the proximity of the Valence band edge to common metals like Au etc. which might make it easier to inject holes compared to electrons owing to a lower Schottky barrier – it might be interesting to study if using low work function metals like Aluminum helps injecting electrons more easily compared to holes.

Te. Thermoelectric power vs. temperature for various acceptor concentrations. Nr. 1...7: $n_a = 6.1 \cdot 10^{14}$; $4.0 \cdot 10^{16}$; $4.8 \cdot 10^{17}$; $6.9 \cdot 10^{17}$; $2.2 \cdot 10^{18}$; $2.5 \cdot 10^{18}$; $5.0 \cdot 10^{18} \text{ cm}^{-3}$ [69G].

References:

- Doukhan, J. C., Drope, R., Farvacque, J. L., Gerlach, E., Grosse, P.: Phys. Status Solidi (b) 64 (1974) 237.
- Rautenberg, M.: Dissertation RWTH Aachen 1977.
- Doukhan, J. C., Farvacque, J. L., in: The Physics of Selenium and Tellurium, ed. by E. Gerlach and P. Grosse, Springer Series in Solid-State Sciences Vol. 13, Springer, Berlin-Heidelberg-New York 1979, p. 126.
- Grosse, P.: Die Festkörpereigenschaften von Tellur, in: Springer Tracts in Modern Physics, Vol. 48, ed. by G. Höhler, Springer, Berlin-Heidelberg-New York 1969
- Baumgart, H. D.: Diplomarbeit, Universität Köln (1966)
- Yuanyue Liu, Wenzhuo Wu, and William A. Goddard, III J. Am. Chem. Soc. 2018, 140, 550–553

5. Table 1: Are WF and IE obtained experimental or calculated values? Also, I suggest giving a schematic drawing to show WF and IE with respect to VBM, CBM and EF.

We thank the reviewer for pointing out this oversight. We apologize for the confusion since we did not clearly mention in the previous version of the manuscript whether these were calculated or experimental values. Both WF and IE values are experimentally extracted from UPS spectra. The procedure (described below) has now been added to the SI to increase the clarity of the manuscript: Ultraviolet photoelectron spectroscopy (UPS) was performed on a Thermo K-Alpha Plus instrument using a He I source (21.2 eV), and values for the work function (WF) and ionization energy (IE) for the Te-PVP and Te-S²⁻ NW film were extracted via UPS spectra. All thin film surfaces were cleaned using an Ar cluster gun (6000 eV/cluster, 150 atoms/cluster, 15 sec) before all UPS experiments. Work functions were extracted as the difference between the UPS radiation source energy and the secondary electron cutoff energy, whereas ionization energies were calculated by measuring the valence band onset.

Since these values strongly depend on the surface of the nanowires and there will be sample inhomogeneities in the amount of sulfur of the surface and leftover PVP (and consequently will drastically affect the work function and ionization energy values), we only use the numbers to demonstrate a qualitative effect of the surface modification on the WF and IE. We find that both values decrease which demonstrate that it is now easy to extract electrons from the surface which is akin to doping a material n-type and the work function moving to lower values is a clear indication that the Fermi level in the film moves towards the conduction band. We corroborate these results with the transistor measurements that also demonstrate a clear trend of the Fermi level shifting towards the conduction band with increased sulfur concentration the surface of the nanowires. Please find below a schematic that shows these trends as a function of doping. We have included the figure in the SI as well.

Figure S24. Cartoon demonstrating the shift in work function due to sulfur doping (work function values are taken from Table 1 in main text). With a band gap of 0.35 eV for bulk Te, the change in work function by 0.3 eV implies that the Fermi level of the S-doped Te NWs lies either in the dopant band or very close to the conduction band of Te.

Changes in the SI:

Ultraviolet photoelectron spectroscopy (UPS) was performed on a Thermo K-Alpha Plus instrument using a He I source (21.2 eV), and values for the work function (WF) and ionization energy (IE) for the Te-PVP and Te-S²⁻ NW film were extracted via UPS spectra. All thin film surfaces were cleaned using an Ar cluster gun (6000 eV/cluster, 150 atoms/cluster, 15 sec) before all UPS experiments. Work functions were extracted as the difference between the UPS radiation source energy and the secondary electron cutoff energy, whereas ionization energies were calculated by measuring the valence band onset.

6. Fig. 4 and relevant analyses are quite confusing and difficult to understand. For example, the definition positive or negative biased VG is not clear. Line 387, page 11, “more importantly, even at the highest positive bias the films are not conductive...”? What does it mean? The former sentence says the film is p-type conductive. Please clearly illustrate this figure and consider moving it to SI since it is only supplemental to the p-n transition.

We thank the Reviewer for their suggestion. Transistors are a robust way to determine and track Fermi level changes in thin films. Since we keep the metal contacts the same and hence the work function of the metal is constant for all the measurements, determining on whether it is easy to inject holes or electrons gives us information regarding the Fermi level in the films with respect to the work function of the metal contact. Now the gate voltage allows us to tune the Fermi level of the films independently and hence provides additional control and hence, additional information regarding the location of the Fermi level with respect to the valence and conduction bands of the thin film material. For example, if we apply a positive bias to the thin film, we move the bands in the semiconductor down in energy which means that it will be now easy to inject electrons in the films. If my semiconductor sample is a n-type material, it will now conduct electrons more easily since if my Fermi level was say at the edge of the conduction band before I applied a positive bias, once I apply the bias, the Fermi level will move deeper into the conduction band – thus leading to more electrons in the conduction band (more density of states as well) and hence more current in the device. However, if my semiconductor is a p-type material and lets say my Fermi level is right at the edge of the valence band, if I apply a positive bias and move the bands down, the Fermi level will be now in the band gap and my current will go down – when we say that even at the highest positive bias, the films are still not conductive, it implies that by applying a positive bias I am in fact moving the Fermi level away from the valence band – which implies that before applying the bias – the Fermi level was closer to the valence band edge. Due to this reason, these set of experiments are a conclusive way to define where the Fermi level in these films lie.

In the thin-film transistor community, it is common practice to demonstrate the transfer and output characteristics of the transistors to justify and prove whether the thin films are p-type, n-type or ambipolar. Hence, we had decided to show all the characteristics in a single figure. As the Reviewer suggested, for the sake of clarity, we have changed the Figure in the main text to make it less dense and included only the Transfer characteristics. We have moved the figure with both the transfer and the output characteristics to the SI instead.

Figure 5. Fermi level shifts with doping. **a**, Schematic cross-section (not to scale) of ion-gel-gated thin-film transistors used to characterize the electrical properties of the doped NWs. The length and width of the channel were $100 \mu\text{m}$ and 2 mm , respectively. Red and blue circles represent positive and negative ions, respectively. **b**, Energy level diagram depicting the relationship between the location of the Fermi energy (E_F), band-edges, doping concentration and corresponding nature of charge carriers. CB and VB refer to the conduction band and the valence band of Te respectively. Transfer characteristics for the **c**, undoped Te-PVP NW sample (with $V_D = 0.1 \text{ V}$) showing p-type transport, **d**, intermediate-doped Te NWs ($\sim 1.5\%$ atomic concentration of sulfur, with $V_D = -0.3 \text{ V}$) showing ambipolar transport. **e**, heavily-doped Te NWs ($\sim 2.4\%$ atomic concentration of sulfur, with $V_D = -1.5 \text{ V}$) showing n-type transport. Black and red curves plot the characteristics on logarithm and linear scales respectively.

Supporting Information:

Figure S26. Fermi level shifts with doping. **a**, Schematic cross-section (not to scale) of ion-gel-gated thin-film transistors used to characterize the electrical properties of the doped NWs. The length and width of the channel were $100\ \mu\text{m}$ and $2\ \text{mm}$, respectively. Red and blue circles represent positive and negative ions, respectively. **b**, Output characteristics showing drain current, I_D , versus the drain voltage, V_D , for undoped Te NWs at various gate voltages (V_G). **c**, Transfer characteristics for the undoped NW sample in **b** with $V_D = 0.1\ \text{V}$. Black and red curves plot the characteristics on logarithm and linear scales respectively. **d**, Energy level diagram depicting the relationship between the location of the Fermi energy (E_F), band-edges, doping concentration and corresponding nature of charge carriers. CB and VB refer to the conduction band and the valence band of Te respectively. **e**, **f**, Output and transfer characteristics (with $V_D = -1.5\ \text{V}$) for heavily-doped Te NWs ($\sim 2.4\%$ atomic concentration of sulfur) showing *n*-type transport **g**, **h**, **i**, Output and transfer characteristics (with $V_D = -0.3\ \text{V}$) for intermediate-doped Te NWs ($\sim 1.5\%$ atomic concentration of sulfur) showing ambipolar transport.

7. The text seems too lengthy with too many subjective or exaggerated expressions. I suggest authors carefully refining the words.

We thank the Referee for their helpful suggestion. We have condensed a few sections. However, we felt it was necessary to include the discussion sections because we are combining a multitude of different approaches (experimental/theoretical/simulations/transistors) in one single report and helps us to emphasize how our approach is different and unique from what is out there. As is evident from the comments, the Reviewers still have a few technical questions regarding the demonstration of the novel doping technique that we presented and hence we feel the discussion is essential to better set the context of our report in relation to other works.

8. More references on Te as thermoelectric material should be cited such as Nature Communications volume 7, Article number: 10287 (2016).

We have incorporated more references as follows:

No prior work – neither in the thin film community nor the bulk community has observed *n*-type transport in this system.⁵⁵⁻⁵⁷

(55) Lin, S.; Li, W.; Chen, Z.; Shen, J.; Ge, B.; Pei, Y. *Nature Communications* 2016, 7, 10287.

(56) Große, P. *Die festkörpereigenschaften von tellur*, Springer: Berlin [u.a.], 1969.

(57) Liu, Y.; Wu, W.; Goddard, W. A. *Journal of the American Chemical Society* **2018**, 140, 550.

Reviewers' Comments:

Reviewer #1:

Remarks to the Author:

The manuscript has certainly been improved, and the STEM/EDX data showing the S distribution on the surface is a particularly strong addition in support of the proposed model. The authors have reasonably addressed many of the major points raised previously. There are still areas that should be addressed, as described in the following comments:

The UPS spectra need to be included in the supporting information and how the valence band onset is determined should be described (for example, the spectra can be fit based on the DFT calculations, a semilog plot can be used to determine the onset, or the onset can be determined based on a linear fit). Currently, the work function and ionization energies measured by UPS are mentioned, but the actual spectra are not shown. It appears that there is also an issue with the interpretation of these spectra and the assignment of the IE. On page 9 the authors state: "Table 1 demonstrates the shift toward lower levels of work function post doping which indicates a shift of the EF from the valence band edge of p-type Te to the conduction band for n-type Te." The values in table 1 do not actually show that Ef shifts from the valence band edge to the conduction band edge, as accompanying the change in the work function is a shift in the ionization energy. In the Te-PVP sample Ef is 0.6 eV from the valence band edge, but in the Te-S2- sample Ef is actually closer to the valence band edge with an energy difference of only 0.4 eV. If the band gap of the material remains constant (or changes by <0.1 eV), then S2- doping would appear to shift the Te sample to be more p-type (i.e., WF close to VB edge). The idea of the material becoming more p-type is not correct based on the Seebeck coefficient, but the separation between WF and IE the data for the two samples does suggest that S2- doping leads to a more p-type material. As another point, the authors state that the band gap of Te is 0.35 eV, which means that in the Te-PVP sample the Fermi energy would be well into the conduction band (0.6 eV difference between Ef and IE). The authors should correct these points and consider how they are analyzing the UPS spectra – which is not currently shown – as it appears the IE values are not correctly interpreted. Additionally, the plot shown in Figure S24 (in S24 the WF is shown at the VB edge, but the IE is listed as 0.6 eV higher than the WF) is not consistent with the UPS, nor is the plot in Figure 5b.

I have concerns about cleaning the Te NW films with Ar ion sputtering prior to UPS measurements, as the sputtering may etch away the thin S2- containing surface layer. XPS results following the Ar ion sputtering should be included to verify that the S composition at the surface is not changing.

The XPS peak that is attributed to electrons from oxidized S 2p electrons is incorrect. The 4s peak of Te falls at a binding energy of 170 eV, which lines up with the peak that is claimed to be from oxidized S. Considering that the peak assigned to oxidized sulfur is reasonably intense in the pure Te-PVP samples, where no sulfur should be present, indicates that this peak is likely from Te and not S. Further support is offered by the EDX data, as no S is observed in the Te-PVP sample (intensity change comparisons and the surface vs. bulk nature of XPS and EDX further support that the 170 eV peak is from Te and not oxidized S). The text should be altered to reflect that this peak is from Te.

The temperature dependence of the electrical conductivity (particularly below 225 K) is odd and the explanation and interpretation is not straightforward. Typically, an increase in the electrical conductivity with decreasing temperature would correlate with decreased scattering and diffusive (delocalized) transport. The authors attribute the observed increase in electrical conductivity with decreasing temperature (below 225 K) to more localized transport. I assume "localized transport" would mean transport through localized states, in which case a hopping mechanism may be inferred. Hopping type transport should show the opposite temperature dependence (electrical conductivity decreases as temperature decreases) though.

In the following sentence the authors refer to section s4 when they should be referring to section

s5. "However, in semiconductors such as Te where the EF lies well within the bandgap, decreasing the charge carrier concentration results in a decrease in the Seebeck coefficient due to a bipolar effect and subsequently leads to a change in its sign once the majority charge carrier inverts (Supplementary Section S4)."

The manuscript would benefit by making the text more concise to improve readability.

Reviewer #2:

Remarks to the Author:

The authors addressed all of my concerns. I recommend publication.

Reviewer #3:

Remarks to the Author:

The revision is satisfied.

Reviewers' comments:

Reviewer #1 (Remarks to the Author):

The manuscript has certainly been improved, and the STEM/EDX data showing the S distribution on the surface is a particularly strong addition in support of the proposed model. The authors have reasonably addressed many of the major points raised previously. There are still areas that should be addressed, as described in the following comments:

We thank the Reviewer for their positive outlook to our changes in the previous version of the manuscript and address their new comments below:

The UPS spectra need to be included in the supporting information and how the valence band onset is determined should be described (for example, the spectra can be fit based on the DFT calculations, a semilog plot can be used to determine the onset, or the onset can be determined based on a linear fit). Currently, the work function and ionization energies measured by UPS are mentioned, but the actual spectra are not shown.

**We have added a Figure in the Supplementary Information to show the UPS spectra and how we obtain the WF values from the same. We use a linear plot to determine the onset – we tried both semilog and linear and they give almost similar values – hence we stuck with a linear plot. Due to issues with IE measurement and extraction (detailed in the next section), we have removed any references to the IE and do not report or use any values in the main text.

Changes incorporated in the SI:

Ultraviolet photoelectron spectroscopy (UPS) was performed on a Thermo K-Alpha Plus instrument using a He I source (21.2 eV) and a -5 V applied bias. Values for the work function (WF) for the Te-PVP and Te-S²⁻ NW film were extracted via UPS spectra. All thin film surfaces were cleaned using an Ar cluster gun (6000 eV/cluster, 150 atoms/cluster, 15 sec) before all UPS experiments. Work functions were extracted as the difference between the UPS radiation source energy and the secondary electron cutoff energy (SECO) using linear fits, whereas ionization energies were calculated by measuring the valence band onset. SECO was determined by fitting the onset curve and finding the point of intersection of that fit line. All data were corrected with respect to the applied bias and the fermi onset energy of a gold standard.

Figure S21. Representative UPS spectra of a set of (a) Te-PVP nanowires and (b) Te-S²⁻ nanowires. The SECO regions are shown in the figures and are used to extract work function (WF) values of 4.04 eV and 3.8 eV for the Te-PVP and the Te-S²⁻ nanowire samples respectively. Please note that the WF values of 4 eV and 3.7 eV reported in the main text are an average from at least 3 measurements on each type of sample.

It appears that there is also an issue with the interpretation of these spectra and the assignment of the IE. On page 9 the authors state: “Table 1 demonstrates the shift toward lower levels of work function post doping which indicates a shift of the EF from the valence band edge of p-type Te to the conduction band for n-type Te.” The values in table 1 do not actually show that E_f shifts from the valence band edge to the conduction band edge, as accompanying the change in the work function is a shift in the ionization energy. In the Te-PVP sample E_f is 0.6 eV from the valence band edge, but in the Te-S²⁻ sample E_f is actually closer to the valence band edge with an energy difference of only 0.4 eV. If the band gap of the material remains constant (or changes by <0.1 eV), then S²⁻-doping would appear to shift the Te sample to be more p-type (i.e., WF close to VB edge).

** We thank the Reviewer for their sharp analysis here. The Reviewer is correct that this analysis isn't clear and agree that some modifications to the text will enhance readability and quality of our work. The key takeaway here is that the work function of the sulfur-doped nanostructures has been decreased, which is consistent with the experimental results showing doping of carriers into the system. Unfortunately, the UPS measurement is not as reliable quantitatively as the other analyses presented in this paper. Our intention is not to provide accurate values for the band edges in the composite systems, but instead to provide a qualitative result to corroborate the suite of analyses in this work in the conclusion that forming Te-S²⁻ nanostructures is associated with the change in band structure and Fermi level shift in this system (transistor measurements/Seebeck coefficient measurements/DFT calculations).

It is important to realize that for the p-type Te nanowires, it is not a thin film of pure Te or bulk Te – what we have are Te nanowires capped with a polymer – PVP. This is a composite organic

inorganic material with a semiconductor/insulator junction. Determining the exact values of IE and work function is non-trivial in these systems as it becomes increasingly hard to determine what is the HOMO of the polymer versus the VBM of the inorganic and what then is the IE of the composite, how do these interacting species influence the measurements and what exactly are we measuring/reporting for the IE values. In the Te-S²⁻ samples, we remove most of the polymer and replace it with S²⁻ (however we still might have some residual PVP on the Te nanowires which can influence the IE values).

We have much more confidence in the work function values because they relate to the Fermi level in the film rather than the ionization energy. The Fermi level in a composite material will equilibrate at the junction of the two species and hence is a good qualitative measure to track changes in the electronic properties – and it is mostly the changes in Fermi level that we are measuring through the transistor and other complementary techniques. As stated earlier, our goal was to simply mention qualitatively that we observe a reduction in the work function values with the surface reengineering. We believe there might be inconsistency in the measured values of IE from the undoped Te samples (with a lot of PVP) and the doped Te samples (with probably some residual PVP).

Similar observation in the WF values of Tellurium have been reported by other groups as well (<https://doi.org/10.1039/C7TA02307C>) where the WF values vary quite a lot (3.6-4.22 eV) simply by changing the surfactant (from PVP to CTAB). Our measured WF values for Te:PVP which is around 4 eV lies close to these reported values. The differing values reported in the literature may arise due to the nature of the surface and the crystallographic orientations. Thus, in our work we use the same Te nanowires with PVP and then without the PVP but with the S²⁻ - thus maintaining the random crystallographic orientation of the nanowires but the only difference being the surface species. In light of these remarks and issues correctly raised by the Referee, we are removing Table 1 on Work function and IE values from the main text, de-emphasizing the result and we are including a new figure in the Supplementary Information that demonstrates how we calculated the WF values from UPS. Please note that the values reported in the main text are an average value from multiple measurements (the analysis of which were done on the UPS/XPS measurement apparatus itself) – we are demonstrating one single analysis in the SI.

The text of the manuscript has been changed to reflect this.

The explanation is further supported by X-ray photoelectron spectroscopy (XPS) measurements (Supplementary Figures S19, S20). A slight blue shift can be observed in the Te peaks in the doped samples which proves that Te atoms are marginally more *p*-type doped in the S-doped samples (Supplementary Figure S20). While isovalent dopants are not perceived as traditional dopants, recent studies have shown that it is indeed possible to dope a material with a corresponding element exhibiting the same valence state. These results are in good agreement with the experimental observations and confirm the proposed mechanism that sulfur adsorption can indeed generate a hybrid *n*-type tellurium-sulfur nanostructure. From the transport measurements wherein we observe *n*-type Te, it is evident that the E_F ought to be close to the conduction band edge. Ultraviolet photoelectron spectroscopy measurements were conducted to measure the work function in Te and Te-S²⁻ thin film samples (Supplementary Figures S21, S22). The measurements

qualitatively demonstrate a shift toward lower levels of work function post doping (from 4.0 eV to 3.7 eV) which, in turn, indicates a shift of the E_F from close to the valence band edge of *p*-type Te to the conduction band for *n*-type Te.

The idea of the material becoming more *p*-type is not correct based on the Seebeck coefficient, but the separation between WF and IE the data for the two samples does suggest that S²⁻ doping leads to a more *p*-type material. As another point, the authors state that the band gap of Te is 0.35 eV, which means that in the Te-PVP sample the Fermi energy would be well into the conduction band (0.6 eV difference between E_f and IE).

**Again, we thank the Reviewer for the opportunity to clarify the work here. Just as in the previous comment, it is important to note that the goal of UPS is not to determine quantitative values for band edges with high accuracy. Instead, the key result is that UPS corroborates the connection between forming Te-S²⁻ composites and the addition of carriers into the system. The Reviewer correctly points out that the band gap of Te is 0.35 eV in bulk systems, but it is well documented that the band gap in nanostructured materials can vary significantly from bulk values. For example, Pradhan et al (<https://pubs.rsc.org/en/content/articlelanding/2017/nr/c7nr01860f#!divAbstract>) cite a 0.65 eV band gap for nanostructured Te. These points have been clarified in the revised text. As described earlier, there might be issues with the measured values of IE due to the presence of the polymer PVP which leads to the discrepancy between the IE and the WF values in Te-PVP sample. Ideally one would expect (since we and everyone else does observe *p*-type behavior in Te-PVP) the IE and the WF values in Te-PVP to be fairly close to one another since the Fermi level would be fairly close to the valence band maximum.

The authors should correct these points and consider how they are analyzing the UPS spectra – which is not currently shown – as it appears the IE values are not correctly interpreted. Additionally, the plot shown in Figure S24 (in S24 the WF is shown at the VB edge, but the IE is listed as 0.6 eV higher than the WF) is not consistent with the UPS, nor is the plot in Figure 5b.

**We thank the Reviewer again for the opportunity to improve the manuscript, and have made changes we believe will address all these comments, as described in the previous points.

I have concerns about cleaning the Te NW films with Ar ion sputtering prior to UPS measurements, as the sputtering may etch away the thin S²⁻ containing surface layer. XPS results following the Ar ion sputtering should be included to verify that the S composition at the surface is not changing.

**Ar ion sputtering should indeed be treated carefully before any UPS/XPS experiments. In this work, we performed many tests to ensure that the Ar cluster sputtering was not significantly altering the surface chemistry of the film. It is important to note that this work featured use of 150 atom/cluster Ar ion clusters. There is a wealth of literature demonstrating that use of large ion clusters is effective in removing surface contaminants while not affecting the chemistry of the nanostructure, relative to monatomic Ar sputtering which has much higher kinetic energy per atom

(https://www.researchgate.net/publication/284132660_Advanced_analysis_tool_for_X-ray_photoelectron_spectroscopy_profiling_Cleaning_of_perovskite_SrTiO3_oxide_surface_using_argon_cluster_ion_source).

Additionally, the nanowires are deposited in a mesh, so the surface sulfur level is present at all depths. Even if we remove some of the sulfur from the wires at the top, the sulfur will persist in all the wires in subsequent layers. Finally, the Ar cluster procedure was validated using a gold standard, so confidence in this method is high.

The XPS peak that is attributed to electrons from oxidized S 2p electrons is incorrect. The 4s peak of Te falls at a binding energy of 170 eV, which lines up with the peak that is claimed to be from oxidized S. Considering that the peak assigned to oxidized sulfur is reasonably intense in the pure Te-PVP samples, where no sulfur should be present, indicates that this peak is likely from Te and not S. Further support is offered by the EDX data, as no S is observed in the Te-PVP sample (intensity change comparisons and the surface vs. bulk nature of XPS and EDX further support that the 170 eV peak is from Te and not oxidized S). The text should be altered to reflect that this peak is from Te.

**We thank the Reviewer for their keen insight on the XPS data. We completely agree with the Referee's interpretation of our data and have incorporated the following changes to alter the text and figure:

Figure S19. High-resolution normalized XPS spectra of undoped Te nanowires (Te-PVP) in black and lightly and heavily doped nanowires (Te-PVP-S²⁻) in red and blue respectively. In the Te-PVP nanowires there is no peak in the spectrum around 162-162.5 eV which suggests the lack of sulfide

species. The presence of a peak around 169-170 eV in undoped Te nanowires corresponds to the Te 4s peak. The apparent red shifting of the peak at 169 eV with increased doping is due to presence of oxidized sulfur species such as sulfates and sulfites which demonstrate S 2p peaks around 169 eV and 166.5 eV respectively leading to a convolution of the Te 4s and S 2p peaks.

The temperature dependence of the electrical conductivity (particularly below 225 K) is odd and the explanation and interpretation is not straightforward. Typically, an increase in the electrical conductivity with decreasing temperature would correlate with decreased scattering and diffusive (delocalized) transport. The authors attribute the observed increase in electrical conductivity with decreasing temperature (below 225 K) to more localized transport. I assume “localized transport” would mean transport through localized states, in which case a hopping mechanism may be inferred. Hopping type transport should show the opposite temperature dependence (electrical conductivity decreases as temperature decreases) though.

**We appreciate the Reviewer’s concern and would like to clarify our wording for the same. When we say localized transport, we do not mean hopping transport through localized states but transport through simply the “localized” dopant band which is isolated from the conduction band from tellurium. The “localized” terminology was used to state that the carriers are restricted within the dopant band and do not access the conduction band. We realize this could have been confusing and hence we remove the term “localized” and alter the main text –

“Combining both results leads us to believe that below 225 K all the carriers are restricted to the dopant band with the Fermi level pinned inside the dopant band (hence the band-like transport) which would intuitively have extremely low mobility due to sparse density of states (hence the low conductivity). Similar Fermi-level pinning has been observed in Ti:PbTe resonant doping.⁵⁵ Above 225 K, carriers gain adequate thermal energy to hop into the conduction band (**Figure 6b**) where they are free to conduct with extremely high mobilities which leads to the sharp conductivity increase (**Figure 6c**)”

One way to think about this is that when the Fermi level is in the dopant band, it is akin to a metal where the Fermi level is within a band rather than in the band gap. Thus, carriers demonstrate “metallic” conductivity or band-like transport instead of activated or Hopping transport, where the conductivity decreases with increasing temperature since carrier concentration remains almost the same while the mobility decreases due to increased carrier scattering. On increasing temperature in our system, these carriers get promoted to the conduction band which is akin to thermally activated carriers getting promoted to the conduction band from the valence band in intrinsic n-type semiconductors, or swapping a lower mobility regime in the dopant band due to less free states to higher mobility regime in the conduction band due to more free states.

Similar results with resonant doping have been reported by Heremans et al. as well for Tl-doped PbTe (Science 2008) - please see figure below - where they first observe an increase in resistivity with increasing temperature (similar to our decrease in conductivity) until about 200 K (where we see our inflection as well) and then a subsequent decrease in resistivity until about 300 K and then again an increase in resistivity at higher temperatures. We might also see this effect once we

promote enough carriers from the dopant band to the conduction band and observe increased scattering.

In the following sentence the authors refer to section s4 when they should be referring to section s5. “However, in semiconductors such as Te where the EF lies well within the bandgap, decreasing the charge carrier concentration results in a decrease in the Seebeck coefficient due to a bipolar effect and subsequently leads to a change in its sign once the majority charge carrier inverts (Supplementary Section S4).”

****We thank the Reviewer for noticing this typo. We have changed it in the main text of the revised version.**

The manuscript would benefit by making the text more concise to improve readability.

****We agree that the manuscript is fairly long. However, in order to address all the Reviewer comments and explain all our observations, we had to expand the text significantly.**

Reviewer #2 (Remarks to the Author):

The authors addressed all of my concerns. I recommend publication.

****We thank the Reviewer for their positive recommendation.**

Reviewer #3 (Remarks to the Author):

The revision is satisfied.

****We thank the Reviewer for their positive recommendation.**

Reviewers' Comments:

Reviewer #1:

Remarks to the Author:

The reviewers have adequately addressed all of my concerns and the work is ready for publication.